

# Event-based analysis of extreme precipitation trends in Italy using hourly convection-permitting reanalyses

Francesco Cavalleri[1,2], Cristian Lussana[3], Francesca Viterbo[2], Michele Brunetti[4], Riccardo Bonanno[2], Veronica Manara[1], Matteo Lacavalla[2], and Maurizio Maugeri[1]

[1]Environmental Science and Policy Department (ESP), University of Milan, Milan, 20133, Italy
[2]Sustainable Development and Energy Resources Department, Research on Electric Systems (RSE), Milan, 20134, Italy
[3]Division for Climate Services, the Norwegian Meteorological Institute, Oslo, 0313, Norway
[4]Institute of Atmospheric Sciences and Climate, National Research Council (CNR-ISAC), Bologna, 40129, Italy

**Correspondence:** Francesco Cavalleri (francesco.cavalleri@unimi.it)

**Abstract.**

The latest generation of high-resolution and convection-permitting reanalyses, capable of representing atmospheric processes at small spatial scales ($\leq$4 km), is crucial for studying the temporal and spatial evolution of phenomena such as convective storms and orographic precipitation. Given the availability of long (>35 years) and continuous convection-permitting reanalysis datasets over Italy, this study investigates the occurrence and characteristics of hourly extreme precipitation events (EPEs) and quantifies their potential increase over time in this region. Using the MERIDA HRES reanalysis (1986–2022), precipitation events are extracted from hourly fields as spatially coherent structures, yielding approximately 160,000 events per year. Each event is characterized by intensity and shape indicators. The resulting HOPE-X (HOurly Precipitation Events and Xtremes) dataset enables a detailed climatological analysis of event frequency, intensity, and spatial scale across seasons. The most extreme component of those events (EPEs), defined based on the mean of local annual maxima in hourly precipitation (RX1hour), show a pronounced increase in occurrence. Specifically, significant upward trends are present during summer in several Alpine and Prealpine regions, as well as in parts of Calabria. In autumn, significant signals emerge in the southern Apennines and in coastal and maritime areas, including the eastern Ligurian coast, eastern Sardinia, the southern Adriatic Sea, and the Ionian Sea. These spatial and seasonal patterns align with regions where convective processes predominantly drive intense, localised precipitation, potentially amplified by climate change. While these findings should be considered in light of known limitations of reanalysis products, such as spatial mismatches with observations and temporal inhomogeneities, multiple independent observational studies support the increase in EPEs during summer and autumn in specific areas. Moreover, the methodology presented here is broadly applicable in any region with access to long-term convection-permitting reanalysis data. In summary, this study offers a contribution to the ongoing discussion on precipitation extremes in Italy and provides guidance for leveraging reanalysis data to enhance infrastructure resilience to short-lived, intense precipitation events.



## 1   Introduction

As global temperatures continue to rise due to climate change (IPCC, 2023), significant alterations in large-scale precipitation patterns are being observed across the globe (Allan et al., 2020). These shifts can trigger even more pronounced changes at the local level (Fowler et al., 2021), particularly in the frequency, intensity, and timing of Extreme Precipitation Events (EPEs). The physical reason for these changes lies in the Clausius-Clapeyron relationship (Hardwick Jones et al., 2010), which describes how a warmer atmosphere can hold more water vapour. Moreover, the rising ocean temperature observed in recent decades (Garcia-Soto et al., 2021) provides more moisture to fill the atmospheric column. The increase in moisture availability produces contrasting effects (Zaitchik et al., 2023): some regions may experience drier conditions, while others may see more intense and frequent rainfall, including extreme precipitation events (EPEs). This effect is generally more pronounced at shorter timescales, such as hourly, than at longer durations (Lenderink et al., 2017). The Mediterranean region, in particular, is recognised as a climate change hotspot, undergoing warming at a faster rate than many other parts of the world (Lionello and Scarascia, 2018). The increasing sea surface temperatures in the Mediterranean contribute to more frequent heavy precipitation events (Senatore et al., 2025), in particular over the Alps and for hourly timescales (Peleg et al., 2025). Within this region, Italy is especially vulnerable to EPEs (Giovannini et al., 2021; Donnini et al., 2023; Padulano et al., 2019), largely due to its complex orography and the interaction between moist air masses, mountain chains, and coastal dynamics (Stocchi and Davolio, 2017; Mazzoglio et al., 2022). All these aspects highlight the need to investigate whether and to what extent climate change is impacting the distribution of hourly precipitation extremes over Italy.

Research on precipitation trends in Italy has been extensive over the past decades, revealing a complex spatial and temporal variability shaped by regional climatic dynamics, topography, and large-scale atmospheric patterns. Several regional investigations based on observational datasets contributed to this discussion, emphasising pronounced local differences. Caloiero et al. (2018, 2021) reported significant negative trends for the period 1951–2016 in both seasonal and annual total rainfall in Southern Italy and inland central regions, especially in winter and autumn. Similarly, in Trentino-Alto Adige (north-eastern Italy), Brugnara et al. (2012) observed a decrease in annual precipitation on the order of 1.0–1.5% per decade in the period 1922-2009, with spring and winter contributing most to the decline. In the same study, the number of wet days significantly decreased east of the Adige Valley (north-western Italy), while trends in extremes (90th, 95th, 99th percentiles) were weak and mostly non-significant. In Tuscany (west-central Italy), Bartolini et al. (2014) found a declining trend in annual rainfall and wet days for the period 1955-2007, largely due to winter and spring decreases. In Calabria (southern Italy), Brunetti et al. (2012), using a high-resolution daily dataset for the period 1923–2006, detected negative trends in mean precipitation intensity (total precipitation per wet day), a reduction in daily precipitation amounts, and a decreased frequency of high-intensity daily events (95th and 99th percentiles). Similarly, Pavan et al. (2019), analysing the ARCIS gridded observational dataset for northern Italy for the period 1961-2015, found summer declines in most regions—driven by fewer rainy days, longer dry spells, and reduced daily intensity—except for the northern Alpine area, which showed increases in both total and intense precipitation. Finally,





Capozzi et al. (2023) analysed multiple stations across Campania (south-western Italy) for the period 2002–2021 and found
an increasing trend in both precipitation intensity and the frequency of heavy rainfall events during autumn, particularly in the
northern part of the region and in mountainous areas.

Collectively, these studies depict a complex evolution of precipitation regimes in Italy, marked by substantial heterogeneity.
However, it is important to note that these studies rely on observational datasets with daily resolution, while the primary
effects of climate change on precipitation are most evident at sub-daily timescales (Lenderink et al., 2017). Indeed, sub-
daily observational datasets typically cover limited regions and relatively short periods, and are generally unavailable for
longer durations while also providing full national coverage (Blenkinsop et al., 2018; Morbidelli et al., 2025). In Italy, the
observational network is extensive and of high quality, but since the 1990s it has been managed at the regional level, resulting
in some heterogeneity among measurement networks. An attempt to homogenize sub-daily observations was made through the
development of the GRIPHO dataset (Fantini, 2019); however, its limited temporal coverage (2001–2016) makes it unsuitable
for long-term trend analysis. Consequently, sub-daily precipitation trends can be investigated using observations available for
specific regions. For example, a delay in the timing of sub-daily rainfall extremes toward autumn was observed in Emilia-
Romagna, along with an overall increase in event magnitude, particularly in the Apennine region (Persiano et al., 2020).
In southern Italy, a growing tendency in hourly extreme rainfall events was observed at several locations, and these trends
generally loose significance over longer durations (Avino et al., 2024). Notably, Mazzoglio et al. (2020) developed a national
dataset of annual daily and sub-daily precipitation maxima for the period 1916 to 2022 (I2-RED), finding that annual maxima
for short durations (particularly 1-hour) have increased nationwide. In contrast, longer durations, such as 24-hour aggregations,
exhibit more spatially variable trends, including some negative tendencies (Mazzoglio et al., 2025). Furthermore, the authors
highlight that the highest quantiles (0.95–0.99) display larger changes than median values. These findings underscore the need
for innovative methodologies to effectively capture and interpret evolving patterns in hourly extreme precipitation across Italy,
beyond observations alone. In fact, rain gauge networks often lack the spatial density required to detect highly localised events,
such as convective storms, unless they occur directly over a station. Conversely, radar and satellite-based measurements, while
offering broader spatial coverage, may suffer from biases during high-intensity events or be affected by terrain-induced signal
blocking. For this reason, convection-permitting reanalyses, blending model outputs with observational data, have proven to be
valuable tools for investigating EPEs and assessing their potential trends over time (Dallan et al., 2024; Poschlod et al., 2021).

In this context, this study aims to investigate the spatial and seasonal characteristics of hourly precipitation events over Italy
and to assess potential changes over time in its most extreme component. To this end, the hourly precipitation fields from the
convection-permitting MEteorological Reanalysis Italian DAtaset – High RESolution, MERIDA HRES (Viterbo et al., 2024)
are employed. This product covers the 37-year period from 1986 to 2022 at about a 4 km resolution. The choice of MERIDA
HRES is supported by previous validation studies that have demonstrated the product's reliability. Its precipitation fields have



been assessed from climatological to daily (Cavalleri et al., 2024a; Viterbo et al., 2024) and hourly (Giordani et al., 2025) timescales. Other studies have inter-compared the performance of many reanalyses over Italy, including MERIDA HRES, highlighting both their strengths and limitations in representing various meteorological variables (Bonanno et al., 2019; Raffa

et al., 2021; Giordani et al., 2023; Cavalleri et al., 2024b). MERIDA HRES has been proven capable of representing convective features of precipitation at fine spatial scales, showing good agreement with both gridded and station-based observations, and demonstrating overall temporal stability when compared with homogenised observational datasets (Cavalleri et al., 2024a). These qualities make it the most appropriate product for hourly precipitation trend analyses. Other convection-permitting models available for Italy, such as MOLOCH (Capecchi et al., 2023) and SPHERA (Cerenzia et al., 2022; Giordani et al., 2023),

have been found to generally produce larger deviations from observed precipitation trends than MERIDA HRES (Cavalleri et al., 2024a), while results from VHR-REA_IT (Raffa et al., 2021) indicate a slightly weaker agreement with daily-scale observations (Cavalleri et al., 2024a). These aspects may limit their applicability for the event-based analysis presented in this study.

Even if convection-permitting reanalyses represent a state-of-the-art, precipitation remains one of the most challenging variables to simulate, and it is not directly assimilated by the reanalyses, but instead derived from assimilated variables such as temperature, pressure, and humidity. These limitations, together with the inherently chaotic nature of the atmosphere, especially at small scales (Hohenegger and Schär, 2007), often lead to some discrepancies between simulated and observed precipitation fields, especially during summer (Cavalleri et al., 2024a). This issue also arises from limitations in the data assim-

ilation frequency (Kalnay et al., 2024) of the driving global reanalyses (e.g., ERA5 assimilates data every 12 hours, much less than typical timescales of convection). While the assimilated observations remain the same regardless of the temporal scale, the sub-daily precipitation fields no longer benefit from the temporal aggregation, which can sometimes hide deficiencies at a smaller scale, making deviations from observations more noticeable. Another relevant aspect is the potential divergence in precipitation trends between observations and reanalyses. Discrepancies in the decadal trend of annual precipitation totals were

highlighted in global reanalyses (Lussana et al., 2024) and Italian regional ones (Cavalleri et al., 2024a). In light of these limitations, an event-based approach has been adopted through the use of a clustering technique. Clustering methods are commonly employed to identify individual precipitation events from gridded datasets, particularly in the context of radar-based observations and operational verification. These techniques typically rely on threshold-based object identification combined with clustering algorithms to isolate spatially coherent precipitation structures. For example, Wernli et al. (2008) describe an object-

based verification method (SAL) that requires the identification of distinct precipitation objects using a threshold proportional to the domain's maximum precipitation value, a strategy also discussed by Davis et al. (2006). Marzban and Sandgathe (2006) provide a broader review of clustering approaches applied to precipitation fields, showing how cluster analysis can be used to define features or objects in both forecast and observation fields, enabling event-based verification. Beyond verification, clustering methods have also been applied to classify sub-daily rainfall events according to their internal structure (Sottile et al.,

2022). Several methods have also been developed to track precipitation events over time (Chang et al., 2016; White et al.,



2017; Li et al., 2020). In this study, however, a straightforward approach to identify precipitation clusters is proposed, based on percentile thresholds that adapt to seasonal variability and the differing precipitation regimes across regions. This methodology does not account for the temporal evolution of the events identified within each cluster but rather focuses on each hourly time step independently, being fully aware of the limitations of this approach, as discussed and acknowledged in the following.


The paper presents in Section 2 the reanalysis dataset used in this study (MERIDA HRES) and describes the methodology adopted to construct the HOurly Precipitation Events and Xtremes (HOPE-X) dataset, publicly available on Zenodo at https://bit.ly/HOPE-X. Section 3 outlines the main results, focusing on the spatial distribution and seasonal patterns of hourly precipitation events, with particular emphasis on the EPEs subset and related trends. Section 4 discusses these findings in the
context of previous studies on precipitation trends and known limitations of reanalysis data. Finally, Section 5 summarises the key conclusions and outlines potential directions for future research.

## 2 Data and Methods

### 2.1 MERIDA HRES, a convection-permitting reanalysis

This study employs the hourly precipitation fields from MERIDA HRES (Viterbo et al., 2024), a reanalysis developed for the
Italian domain, resolving explicit convection to better represent localised and intense precipitation events. MERIDA HRES, developed by Ricerca sul Sistema Energetico (RSE), employs the Weather Research and Forecasting (WRF) model to dynamically downscale the global ERA5 reanalysis (Hersbach et al., 2020) to a high-resolution grid of approximately 4 km over Italy. It is driven by large-scale initial and boundary conditions from ERA5 and applies a spectral nudging technique (von Storch et al., 2000) to constrain synoptic-scale features while filtering out smaller-scale perturbations that could introduce spurious
signals. Additionally, SYNOP surface air temperature observations are assimilated through an observational nudging technique (Liu et al., 2012; Bonanno et al., 2019; Viterbo et al., 2024), further enhancing the representation of regional atmospheric characteristics. The dataset spans the period from 1986 to 2022, but is constantly updated with about a 2-year lag. The analyses for this work are calculated over the domain 5.84°E to 18.96°E in longitude and 35.37°N to 48.25°N, centred on the Italian peninsula, for the period 1986-2022, enclosing the full period of availability for MERIDA HRES.

### 2.2 Event detection and characterization

The event-detection method used in this study aims to identify spatially coherent precipitation events from each hourly field. It applies the 50th percentile of precipitation values exceeding 1 mm as a threshold, computed for each grid point of the MERIDA HRES reanalysis. Thresholds are calculated separately for each season. Finally, a spatial smoothing filter with a 20 km radius is applied to reduce noise and improve spatial consistency across neighbouring grid cells (Figure 1).





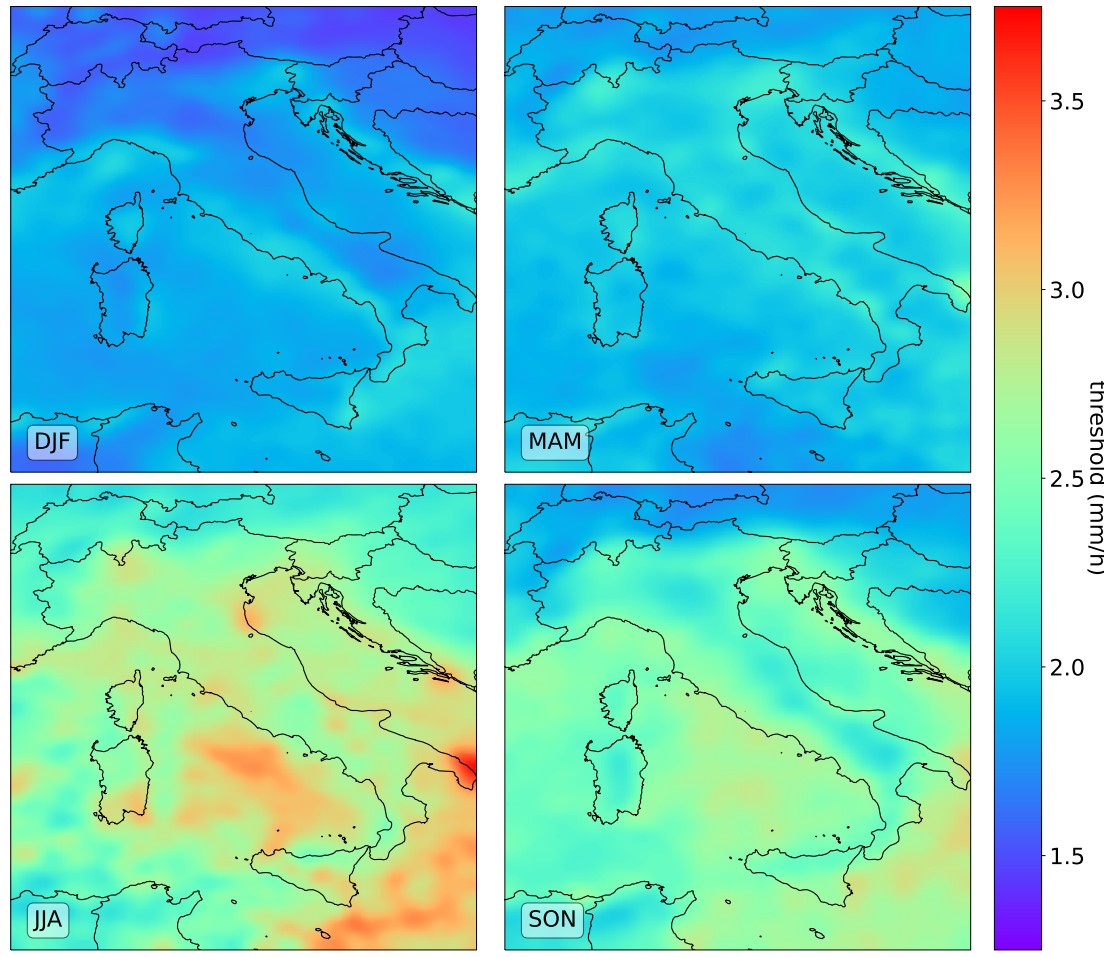

**Figure 1.** Seasonal maps of the 50th percentile of hourly precipitation values above 1 mm, used as clustering thresholds.

Precipitation values below 1 mm/h are excluded to distinguish meaningful precipitation from background noise. Below this value, spatial variability is very high, whereas it significantly decreases above it, indicating that precipitation becomes more spatially coherent and representative of broader areas (Lussana et al., 2023). At the beginning, a fixed 1 mm threshold was applied to detect hourly precipitation events. Nevertheless, the choice of a uniform threshold across the entire domain and for all seasons did not adequately account for the spatial and seasonal variability of precipitation regimes, leading to the merging

of multiple distinct convective cells into a single, large cluster that did not reflect the localised nature of these events. This mismatch between the actual physical scale of convective systems and the scale of the detected clusters motivated the choice of a percentile-based threshold, computed using all hourly precipitation values greater than 1 mm, separately for each season. The thresholds, defined as the 50th percentile of precipitation above 1 mm (Figure 1), are applied to all hourly precipitation fields from 1986 to 2022. Contiguous grid points exceeding these thresholds are identified as an individual event. To reduce





noise, clusters composed of fewer than five grid points are excluded: approximately 95% of them exhibit intensities below 10 mm/h, and therefore have a negligible impact on the focus of this study on extreme precipitation. Retained clusters are identified as individual relevant precipitation events. More specifically, 'relevant' refers to spatially continuous precipitation structures that occur in more than half of the instances within a given area and season, and are therefore considered sufficient to identify detectable, and at times significant, events. Hereafter, the term 'event' denotes the precipitation structures identified

using this method. Figure 2 shows an example of the event-detection procedure, applied to the hourly precipitation field of 20 October 2011 at 13:00:00 UTC. On that day, intense precipitation affected Rome and the surrounding areas, causing several floods throughout the city and widespread power outages (Bonanno et al., 2019).

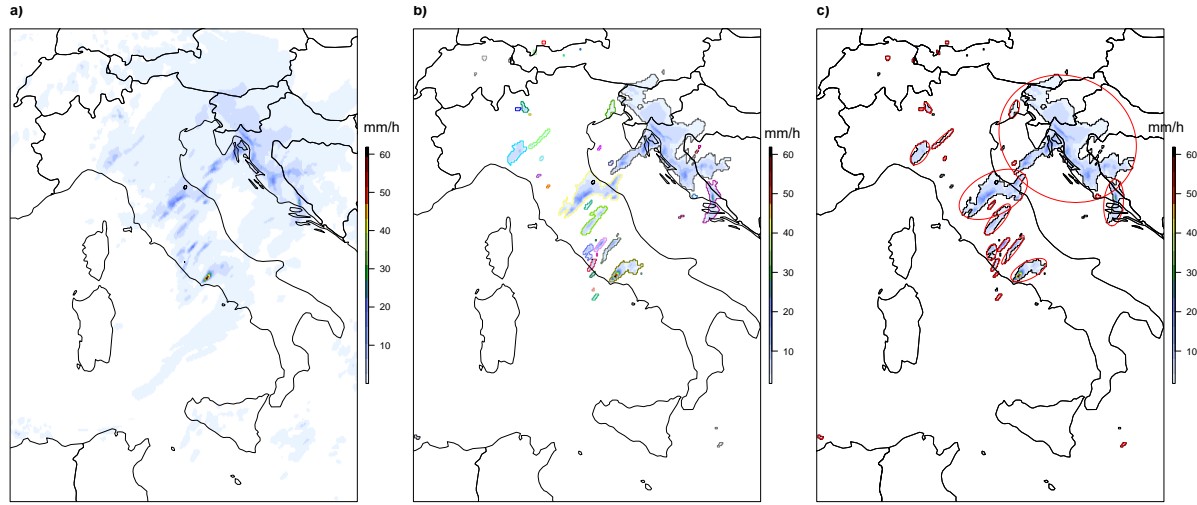

**Figure 2.** Example of event selection process for the day 20th October 2011, 13:00:00 UTC. a) raw precipitation field, b) after applying the threshold and the clustering (each border colour represents a different cluster), c) minimum enclosing ellipses (in red) identify retained events.

For each identified event, an enclosing ellipse is calculated following the methodology of Wernli et al. (2008), and key proper-

ties are extracted. The characteristics and methods used for their calculation are summarised and explained in Table 1. The table presents only the characteristics directly used in this study; however, many additional variables are included in the complete database.

## 2.3 Event-based statistics

The event detection methodology explained above resulted in the HOPE-X dataset. Several indicators are computed from

this dataset. First, seasonal distributions of selected characteristics listed in Table 1 are obtained to provide an overview of




**Table 1.** The characteristics recorded for each event which are relevant for this study.

| Variable Name | Description and/or definition |
|---|---|
| time | Date and hour of the field where the object is detected. |
| tp_max | Maximum total precipitation value within the object. |
| lon_max | Longitude where maximum precipitation (tp_max) occurs. |
| lat_max | Latitude where maximum precipitation (tp_max) occurs. |
| lon_wavg | Intensity-weighted average longitude of the object. |
| lat_wavg | Intensity-weighted average latitude of the object. |
| area | Area of the object (number of grid cells). |
| tot_tp | Total precipitation summed over the entire object area. |
| axis_maj | Length of the major axis of the object (in degrees). |

their values. In particular, the average intensity (tot_tp/area), peak intensity (tp_max), and spatial scale (axis_maj) of individual events are examined. Then, spatial patterns of hourly precipitation are investigated, accounting for location uncertainty inherent in reanalysis data. To this end, a spatial aggregation is applied using a moving window of 0.5° with 0.1° increments in both latitude and longitude. In each of these windows, the number of events (N) whose centre of mass (lat_wavg,lon_wavg) fell inside the window is counted. Because the sliding distance (0.1°) is smaller than the window size (0.5°), a single event is counted in multiple adjacent windows, ensuring smooth spatial transitions. The Average Intensity (AvIn), the Peak Intensity (PkIn), and Spatial Scale (SpS) are obtained by averaging over the events within each window tot_tp/area, tp_max and axis_maj respectively, as detailed in Table 2.

**Table 2.** Description of indicators used for the analyses and their units of measure, where $t = 1,\ldots,T$ indicates the different hours within the period over which the indicator is computed (typically a season), and $m = 1,\ldots,M$ denotes the different events occurring at a timestep within a given spatial window.

| Short Name | Mathematical description | Unit of Measure |
|---|---|---|
| N | $\sum_{t=1}^{T}\sum_{m=1}^{M}1$ | number |
| AvIn | $\frac{1}{T}\sum_{t=1}^{T}\frac{1}{M}\sum_{m=1}^{M}$ tot_tp$_m$/area$_m$. | mm/h |
| PkIn | $\frac{1}{T}\sum_{t=1}^{T}\frac{1}{M}\sum_{m=1}^{M}$ tp_max$_m$ | mm/h |
| SpS | $\frac{1}{T}\sum_{t=1}^{T}\frac{1}{M}\sum_{m=1}^{M}$ axis_maj$_m$ | km |

Both AvIn and PkIn are expressed in millimetres per hour (mm/h), but they reflect different aspects of precipitation intensity. AvIn represents the average intensity across all grid points of all events within a given window, while PkIn refers to the mean of the maximum intensities recorded at a single point for each event. SpS indicates the average maximum linear extent of the




events within the same window. Finally, these values are cumulated (for N) or averaged (for AvIn, PkIn, SpS) over time to obtain aggregated values for each location. Statistics on the full event dataset, including climatologies of these indicators, are presented in Section 3.1.

## 2.4 Extreme precipitation events sub-setting

After characterising the properties of the hourly precipitation events, the extremes are subset from the full event dataset. The selection criterion is based on the average of the annual maxima of hourly precipitation (RX1hour), calculated for each grid point and each year. The resulting time series of 37 RX1hour values are then averaged throughout the period 1986-2022 to derive a threshold value for each cell of the grid, representing the average RX1hour at that location. This approach is similar to the methodology used by Lavers et al. (2025), who introduced the Extreme Rain Multiplier (ERM) to classify extreme daily precipitation events. Lavers et al. (2025) employ ERA5, which is a global reanalysis product with a 0.25° grid spacing, and consider daily precipitation accumulations to compute the mean of the annual daily maxima (RX1day). The daily accumulation is the most appropriate timescale considering the coarse spatial scale of ERA5 (Chinita et al., 2022; Raffa et al., 2021). In contrast, this study uses a regional convection-permitting reanalysis, which provides a more accurate representation of hourly precipitation and associated extremes. Therefore, it is possible to define a threshold based on hourly maxima (RX1hour). As the last step, a Gaussian filter with a 20 km radius is applied to smooth the average RX1hour field and reduce local-scale noise (Figure 3).

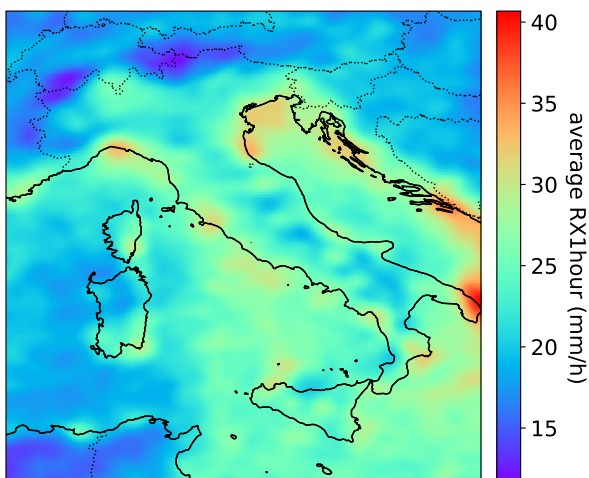

**Figure 3.** 1986-2022 average of the annual maxima of hourly precipitation (RX1hour), after the application of a 20 km Gaussian filter.

In deciding the most suitable filtering radius, several values were tested. Radii larger than 20 km excessively smooth areas with higher thresholds, reducing the ability to resolve localised extremes. Conversely, smaller radii retained too much noise, limiting the effectiveness of the filtering in isolating coherent precipitation structures. Additionally, the 20 km scale corresponds to the boundary between the meso-$\beta$ and meso-$\gamma$ atmospheric scales (Thunis and Bornstein, 1996), below which convective





events typically occur. Finally, a precipitation event is identified as EPE if its maximum precipitation value (`tp_max`) exceeds the average RX1hour value in the position where it occurred (`lat_max,lon_max`).

## 2.5 Extreme Precipitation Event statistics and trends

Extreme statistics are calculated within the subset of events classified as EPEs with the same methodology described in Section 2.3. Subsequently, the trends of the yearly series of N, AvIn, PkIn and SpS are computed (results shown in Section 3.2). On each moving window, the trend analysis is performed using the Theil–Sen slope estimator (Sen, 1968), suitable for non-parametric data. The statistical significance of the trends is evaluated using the Mann–Kendall test (Mann, 1945; McLeod, 2005). To control for the multiple testing problem across the spatial domain, the False Discovery Rate (FDR) correction is

applied (Benjamini and Hochberg, 1995; Wilks, 2006). Since the FDR procedure tends to be conservative in the presence of spatial correlation, approximately correct global results can be obtained by setting the FDR threshold to twice the desired global significance level (Wilks, 2016, 2019). Therefore, results are considered statistically significant if the FDR-corrected p-value is below 0.1, corresponding to a global significance level of 0.05. The results of the trend analysis are presented in the Results section 3.3.

# 3 Results

## 3.1 Full precipitation events dataset analyses

The HOPE-X dataset consists of approximately 160.000 precipitation events per year over the period 1986-2022. The interannual variability, calculated as the relative standard deviation of the annual number of events, is around 10%. At the seasonal level, the highest number of events is generally recorded in autumn (SON), accounting for 29% of the total, while summer

(JJA) shows the lowest share, with 21%. Winter (DJF) and spring (MAM) contribute similarly, representing 26% and 24% of the total number of events, respectively. The fraction of hours showing no identified events across the entire domain varies seasonally, with approximately 11% for winter, 12% for spring, 9% for summer, and 7% for autumn. The number of events detected per hour follows a distribution that decreases with increasing event count (Figure 4).

The maximum number of events recorded in a single hour is 136, observed at 14:00 on June 11, 1992, as a result of a

widespread low-pressure area influencing the whole Italian peninsula. Intensity and spatial scale distributions exhibit markedly skewed shapes, with a sharp peak at low values followed by an approximately exponential decay as their magnitude increases (Figure 5). During summer and autumn, precipitation events tend to exhibit higher median values and heavier tails for both `tot_tp/area` (Figure 5a) and `tp_max` (Figure 5b). The `axis_maj` (Figure 5c) distributions show less pronounced seasonal variation, with only summer displaying slightly smaller-scale events. A small percentage of events fall outside the range

of the distributions plotted in Figure 5: 0.22% of events exhibit an `tot_tp/area` greater than 15 mm/h, 0.96% have a `tp_max` above 40 mm/h, and 2.98% show a `axis_maj` larger than 100 km. According to definitions of atmospheric scales in the scientific literature (Thunis and Bornstein, 1996), phenomena with lifetimes ranging from about one hour to one day—such





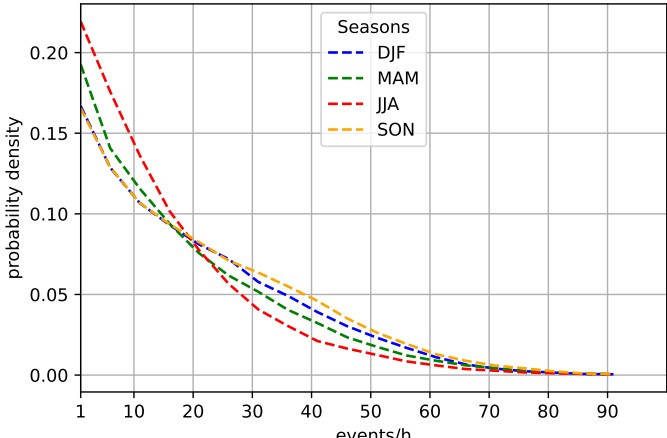

**Figure 4.** Distribution of the number of events recorded per hour. Values are normalised by the total number of hours in each season (24 × 90 × 37). Bin width: 5.

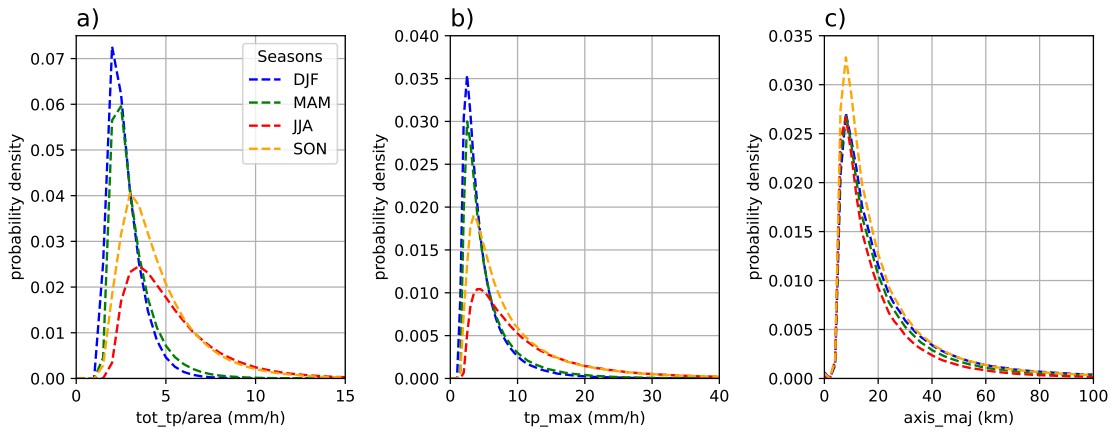

**Figure 5.** Seasonal distributions of a) average intensity, b) peak intensity, and c) spatial scale of precipitation events. Bin width: 0.5 mm for intensity variables, 2 km for spatial scale. Distributions are normalised by the total number of events; that is, the sum of the integral of the four seasonal distributions gives 1.

as isolated thunderstorms or groups of storms—typically occur within the lower portion of the mesoscale, with spatial extents from approximately 1–2 km up to 200 km. Our results confirm that, at the hourly timescale, significant precipitation events generally fall within the meso-$\gamma$ scale (2–20 km), with only occasional instances extending to larger spatial scales. This finding is particularly relevant for applications that require knowledge of the typical spatial scales of hourly precipitation events, such as spatial analysis of precipitation fields (Fortin et al., 2018; Van Hyfte et al., 2023). Moreover, it is important to note that, in




this dataset, high intensities generally correspond to smaller spatial scales (see Supplementary Material, Figure S1). Overall, the majority of events concentrate on low values of intensity and spatial scale. This underscores the need to isolate the most

extreme events to better understand their specific characteristics.

The results on spatial and seasonal distribution of events is analysed using the methodology described in Section 2.3, resulting in seasonal climatological means of N (Figure 6), SpS (Figure 7), AvIn (Figure 8) and PkIn (Figure 9). Higher values in a given area indicate a greater number of events with their centre of mass located within that region (for N), or larger values of AvIn, PkIn and SpS for those same events. Events may extend beyond the boundaries of the window in which they

are counted, since the averaging considers only the events whose centre of mass lies within the window. However, most of the recorded events are well delimited in a small space (Figure 5c). It is also important to emphasise that these means are computed from distributions that are strongly right-skewed, as shown in Figure 5. Consequently, the values presented in the maps should be interpreted with some caution. While they may not fully capture the absolute characteristics of typical event occurrences, scales, and intensities, they remain informative when used to explore spatial and seasonal patterns and relative

differences across regions.

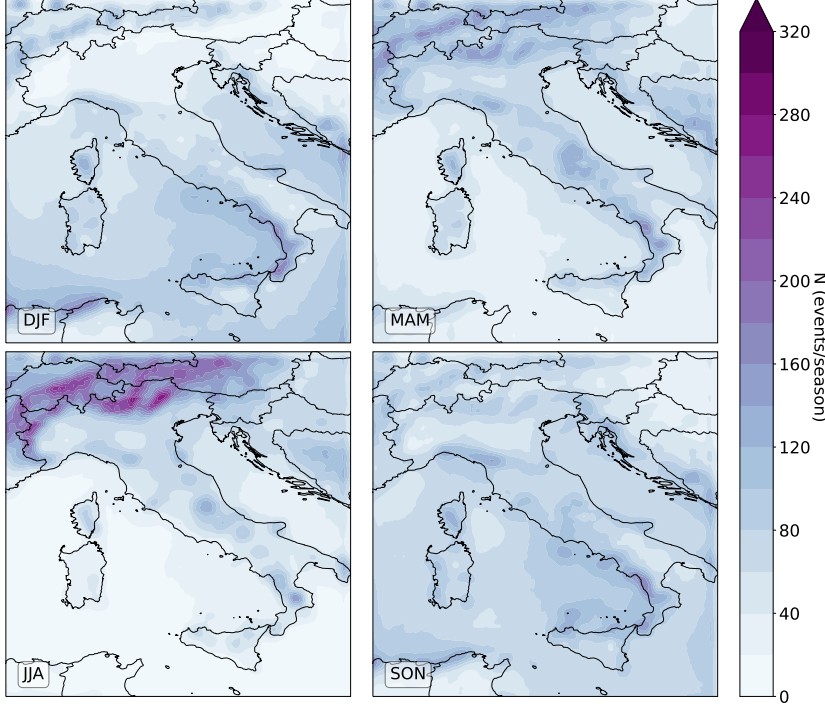

**Figure 6.** Seasonal map of N occurring within the $0.5 \times 0.5$ windows (step size 0.1), averaged over the period 1986–2022.

The spatial distribution of N (Figure 6) shows that, in summer, most of the events occur in the Prealpine regions, with secondary hotspots along parts of the Apennines, and almost no events over the sea. In autumn and winter, the areas with high N shift toward coastal and offshore areas, particularly along the Tyrrhenian and Ligurian seas. During spring, the Prealps and





Apennines are again prominent, although the occurrences are generally lower than in summer. The Po Valley and Prealpine
region exhibit very low N of events during the winter season. These seasonal patterns reflect the typical climatology of con-
vective precipitation in Italy, which tends to be more frequent during the warmer months and over mountainous regions and
coastal areas (Lombardo and Bitting, 2024).

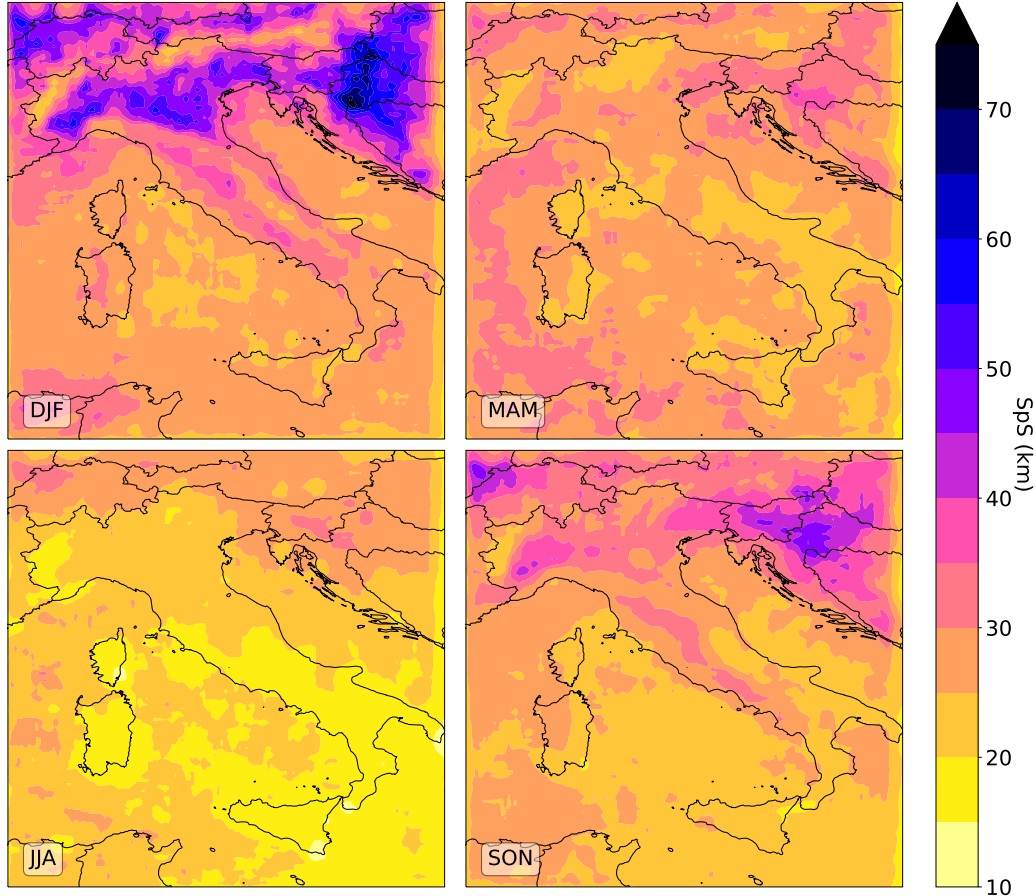

**Figure 7.** Seasonal map of SpS of the events occurring within the $0.5 \times 0.5$ window (step size 0.1), averaged annually over the period
1986–2022.

The seasonal maps of SpS (Figure 7) reveal that during summer events are generally smaller, with typical average SpS ranging
between 10 and 20 km, especially along coastal areas and in southern Italy, and from 20 to 30 km in the other Italian areas.
This is consistent with the convective nature of summer precipitation. Springs show slightly larger SpS, but still below 30
km over the Prealps and in southern regions, where autumn also displays those values of SpS, despite showing larger ones
over plain areas in the north and central Italy. In contrast, winter is characterised by generally larger events, especially over




the Po Plain, where average spatial scales commonly reach 50 km, exceeding values registered over the Alps and Apennines.
This broader spatial extent reflects the influence of large-scale synoptic systems typical of wintertime precipitations over Italy.
Overall, these patterns highlight a seasonal modulation in SpS, reflecting the shift from localised convective activity in summer
to more widespread, synoptic-driven precipitation in autumn and winter.

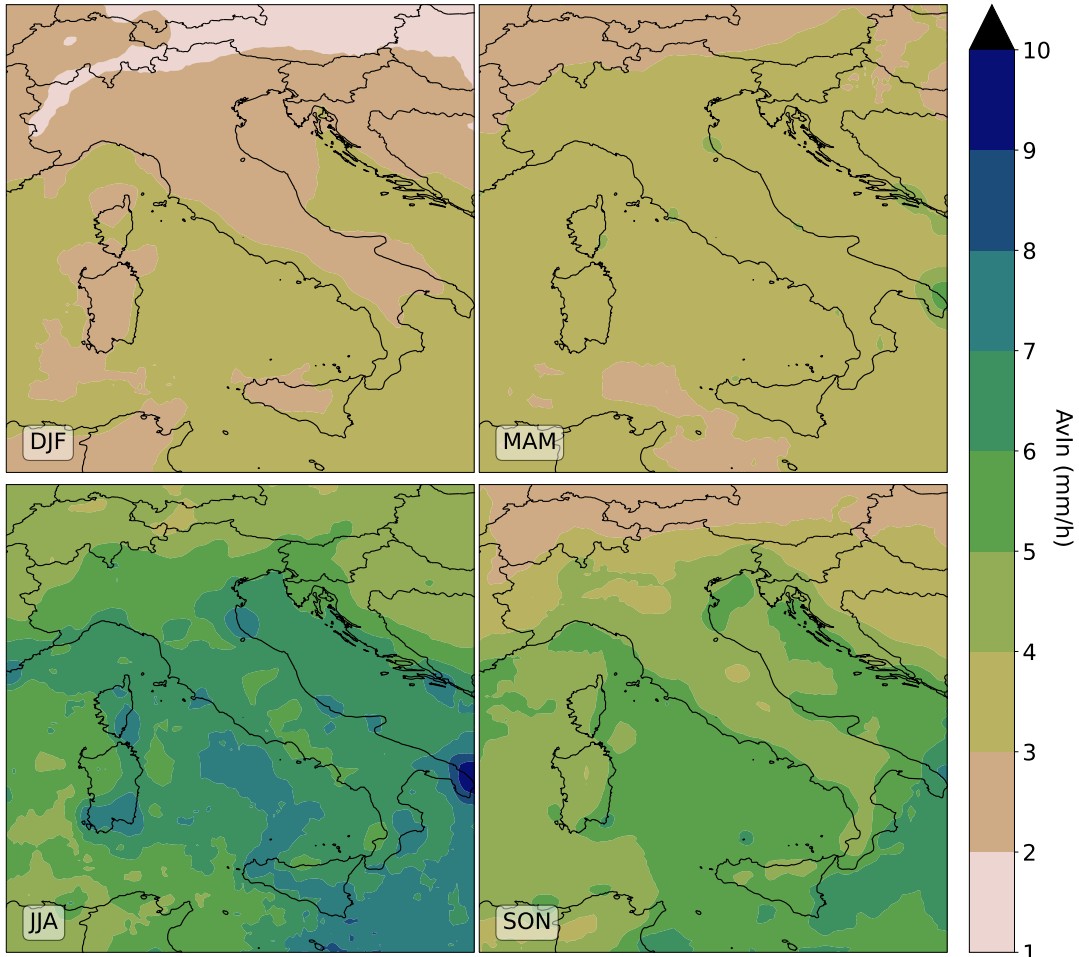

**Figure 8.** Seasonal map of AvIn of the events occurring within the $0.5 \times 0.5$ window (step size 0.1), averaged annually over the period
1986–2022.

The spatial distribution of AvIn (Figure 8) highlights summer as the season with the highest average intensities, often
exceeding 5 mm/h with maxima of more than 7 mm/h in some areas along the Adriatic coast, such as Calabria, the Tyrrenian
sea, southeastern parts of the islands and southern Apulia. In winter, intensities generally range between 2 and 3 mm/h over
most of the peninsula, dropping below 2 mm/h along the Alpine arc and exceeding this value only slightly in some southern
areas and along the Tyrrhenian coast. During spring, values between 3 and 4 mm/h are widespread throughout Italy, except for



isolated spots over 4 mm/h in similar areas to those observed in summer. In autumn, slightly higher intensities, ranging from 4 to 5 mm/h, cover most of the country, while lower values persist only in the Prealpine and Alpine regions. Intensities above 5 mm/h are found mainly along the coastal areas and over the surrounding seas.

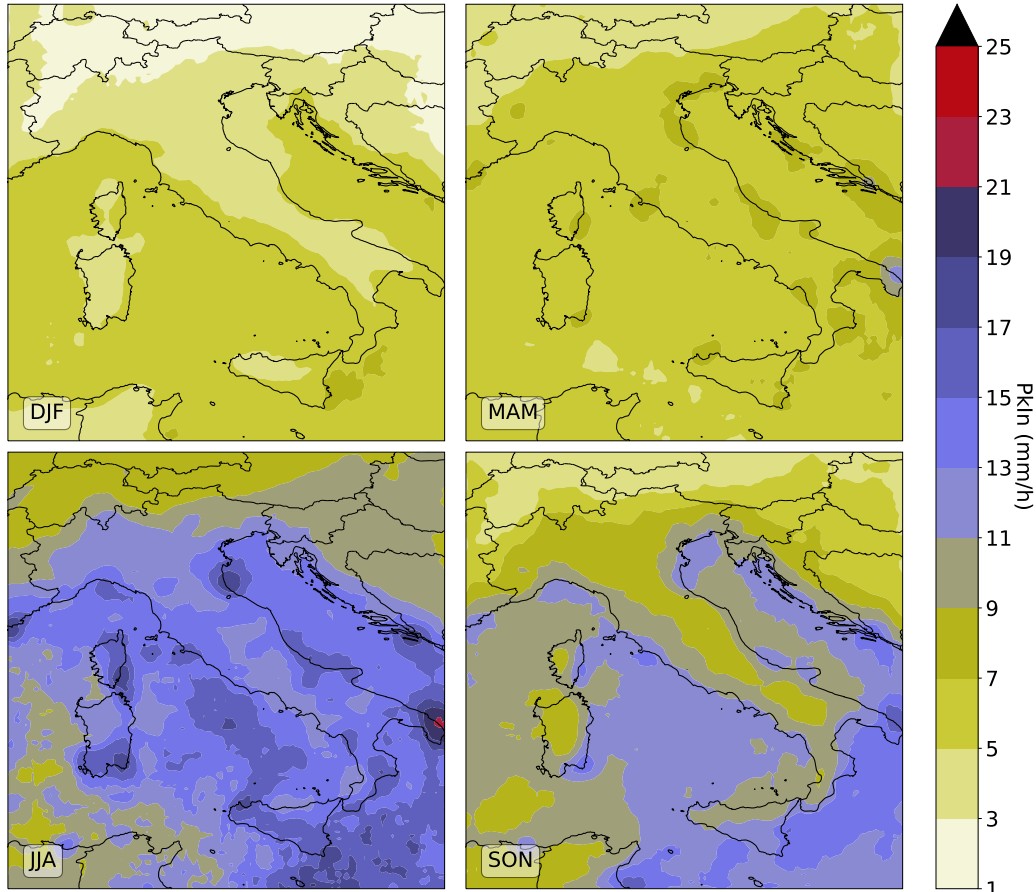

**Figure 9.** Seasonal map of PkIn of the events occurring within the 0.5 × 0.5 window (step size 0.1), averaged annually over the period 1986–2022.

The spatial distribution of local PkIn (Figure 9) further emphasises the seasonal contrasts. These maps closely resemble those for AvIn, although PkIn are generally higher. PkIn increases from winter values ranging between 2 and 7 mm/h to well over 15 mm/h during summer, with spring and autumn showing intermediate values. Notably, in autumn, PkIn exceeding 10 mm/h are mostly confined to coastal areas and the surrounding seas. In summer, PkIn surpasses 17 mm/h in the same regions characterised by high summer AvIn (Figure 8).

Since it is not straightforward to determine to what extent the seasonal differences in Figures 6, 7, 8, and 9 are influenced by the use of seasonally varying thresholds for event selection, a set of corresponding figures derived from the event-based dataset built using a fixed 1 mm threshold is provided in the Supplementary Material (Figures S2, S3, S4, and S5). These figures display



very similar spatial patterns—albeit with generally lower intensity values—suggesting that the observed seasonal differences
primarily reflect genuine variability rather than artefacts introduced by the clustering method. Overall, the climatological
maps of hourly precipitation event indicators are consistent with the established climatology of the region (Crespi et al.,
2018; Giordani et al., 2025). However, while AvIn and PkIn (Figures 8 and 9) appropriately reflect higher values during the
autumn and summer seasons, they also display certain inconsistencies. In particular, some areas exhibit an overrepresentation
of convective activity during summer, which may not fully align with observed patterns. This issue will be examined in greater
detail in the Discussion section 4.

## 3.2 Extreme Precipitation Events analyses

To gain insight into the most intense precipitation fraction, a focused analysis is conducted on a subset of EPEs, selected
according to the criterion detailed in Section 2.4. The filtering procedure resulted in approximately 4.8% of all events as
EPEs, corresponding to an average of around 7800 hourly events per year across the whole domain, with a notable interannual
variability of about 30%. Most EPEs are selected from summer (11% of all summer events) and autumn (7%), while only a
marginal fraction are identified in spring (1.5%) and winter (0.5%). This seasonal breakdown results from the combined effect
of higher thresholds applied for event identification during summer (Figure 1), which selected relatively intense precipitation
events even within the full dataset for that season, and the use of a fixed threshold (average RX1hour) for EPE selection
throughout the year. The greater number of hourly EPEs in summer and autumn is also consistent with the expectation that
hourly precipitation more effectively captures extremes and their associated impacts at smaller spatial scales, such as convective
storms and other meso-$\gamma$ scale phenomena, particularly prevalent during the warmer seasons. Consequently, the Sections 3.2
and 3.3 focus exclusively on summer and autumn precipitation extremes.

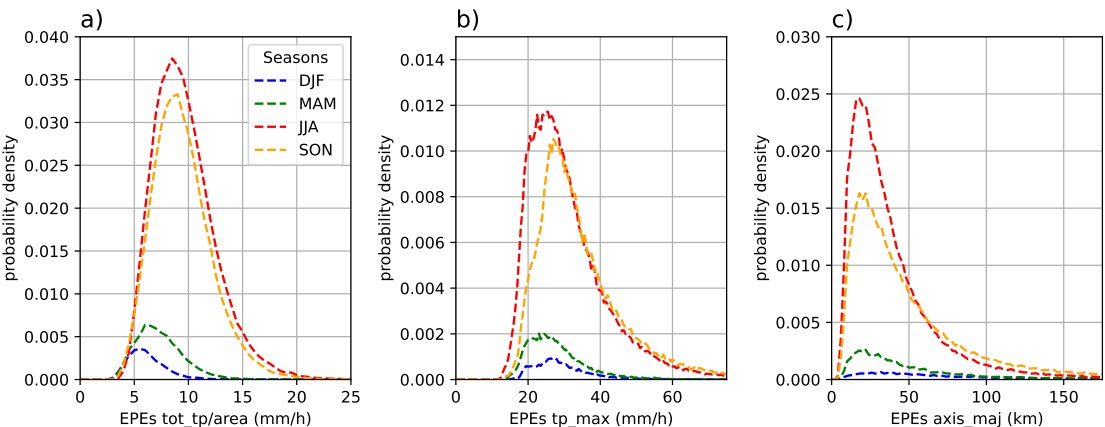

**Figure 10.** Seasonal distributions within the EPEs subset: a) average intensity, b) peak intensity, and c) spatial scale of EPEs. Distributions
are normalised by the total number of EPEs; that is, the sum of the integral of the four seasonal distributions gives 1. Binning as in Figure 5.



A comparison between the distributions of intensity and spatial scale within the EPE subset (Figure 10) and those from the full set of events (Figure 5) confirms that the applied filter effectively excludes a substantial number of events from the lower tails of the distributions. This effect is quite obvious for the peak intensity, which is explicitly used as the filtering parameter. However, it also significantly influences the distribution of average intensity, suggesting that, on average, EPEs are not only more intense locally but also tend to have higher average values. Moreover, the peaks of the spatial scale distributions are shifted towards larger values. Summarising, the applied filtering leads to the exclusion of a large fraction of small and weak events, not meaningful for the EPEs analysis.

The climatological seasonal maps of N within the EPEs subset (Figure 11) highlights clear seasonal differences between summer and autumn.

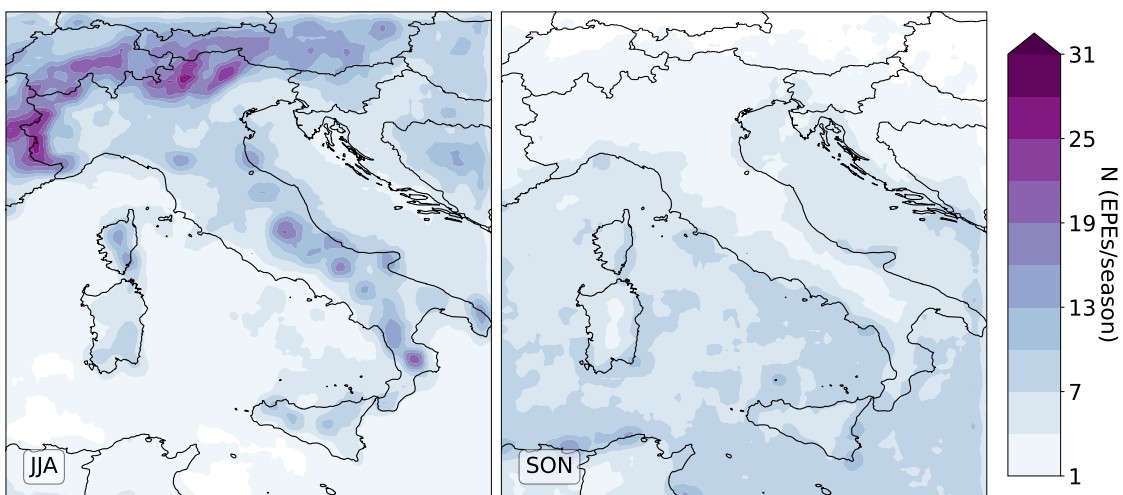

**Figure 11.** JJA and SON maps of number (N) of EPEs occurring within the 0.5 × 0.5 window (step size 0.1), averaged annually over the period 1986–2022.

In summer, EPEs occur predominantly over mountainous areas, particularly the Alps and some spots along the Apennines, and Calabria, reaching 20 to 30 events per 0.5° grid window per year. In contrast, coastal and marine regions display a significantly lower N, often fewer than 3 per window. In autumn, N is substantially less compared to summer. However, a clear spatial shift emerges: mountain areas experience fewer to none events, while coastal and marine zones see some, with over 7 occurrences per window observed along many stretches of coastline. This seasonal redistribution is likely driven by the persistence of summer-like convective activity into early autumn at lower latitudes, where warm sea surface temperatures continue to support intense storm development (Cheng et al., 2022; Argüeso et al., 2024).

Marked differences between summer and autumn also emerge in terms of SpS (Figure 12). In summer, EPEs rarely exceed 50 km in size, except in limited areas such as Friuli (North-East) and South Switzerland, and remain well below 20 km across much of southern Italy and the islands. Conversely, in autumn, significantly larger events (exceeding 100 km in spatial extent)



are frequently observed. Spatial scales remain smaller mainly in the south, along the Adriatic coast, and over the islands. This suggests that EPEs are typically small, convective systems during summer across most of the Italian territory, and during autumn along the southern coastlines. In contrast, in northern Italy and neighbouring regions, autumn EPEs are more frequently associated with larger-scale systems.

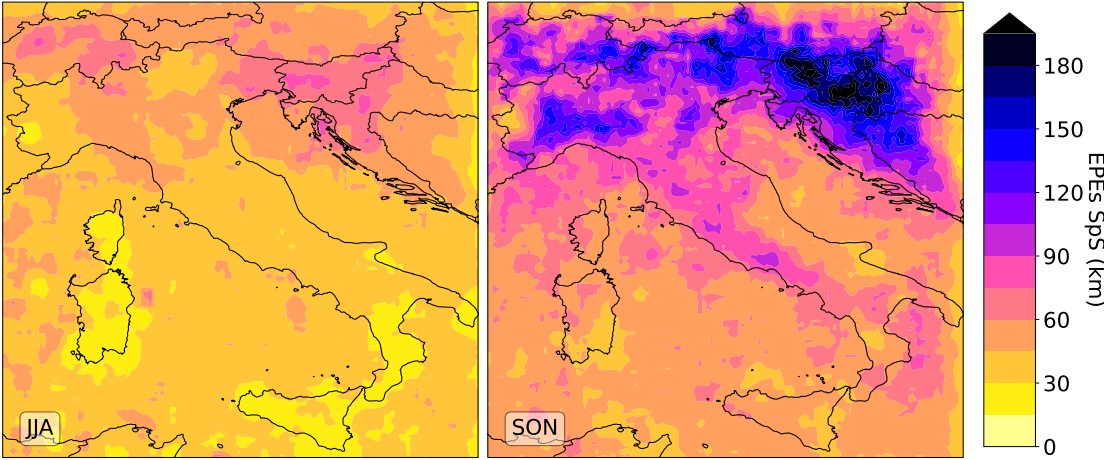

**Figure 12.** JJA and SON maps of the Spatial Scale (SpS) of EPEs occurring within the $0.5 \times 0.5$ window (step size 0.1), averaged annually over the period 1986–2022.

The climatological maps for the AvIn and the PkIn of EPEs are provided in the Supplementary Material (Figures S6 and S7). Overall, their spatial patterns closely resemble those observed for the full set of precipitation events, though with generally higher values, due to the filtering, which also reduces the seasonal differences. Specifically, the AvIn range from approximately 5 to 15 mm/h, increasing from the Alpine regions to southern Italy for both seasons, while the PkIn range from 20 up to 50 mm/h, with the lowest values again found over the Alps and the highest values concentrated in the same hotspots highlighted before, such as the southern Apulia.

### 3.3 Extreme Precipitation Event trends

Finally, given the context provided by the previous results, a trend analysis within the subset of EPEs is conducted, following the methodology outlined in Section 2.5. Significant trends in the number of EPEs occurrences (N) during summer and autumn are detected (Figure 13). Trends are expressed as percentages relative to the seasonal and local mean values of N (i.e., normalised by the values shown in Figure 11). For example, a 10% trend in Figure 13 means a decadal increase of 10% in N, indicating that, on average in that area, approximately 30% more EPEs occur at the end of the study period compared to its beginning. Overall, a general increase in EPEs occurrences is detected across the peninsula, even though only some regions exhibit statistically significant trends. In summer, a significant increase of approximately 20% to 30% per decade is detected across several Alpine and Prealpine regions, and in some parts of Calabria. In autumn, significant trends are primarily concentrated over the southern



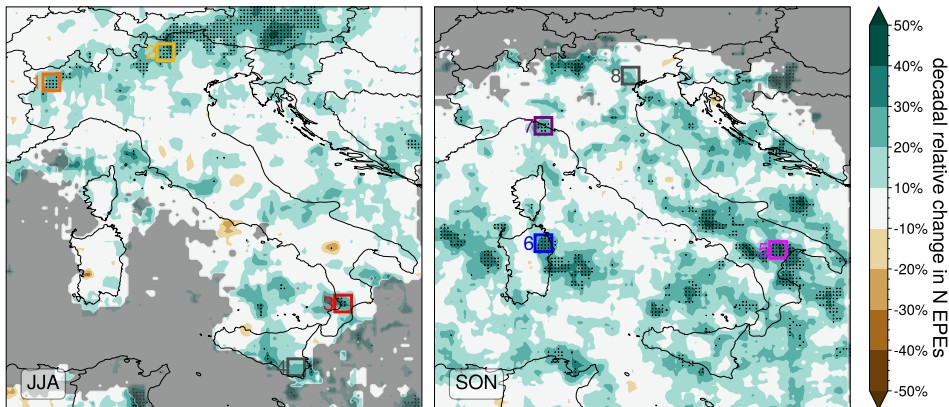

**Figure 13.** Maps of the significant decadal relative trends in the number of Extreme Precipitation Events (EPEs) occurring within each 0.5° × 0.5° window (sliding step: 0.1°) for summer (JJA) and autumn (SON). Black dots indicate statistically significant trends. Areas with more than 10 years without EPEs are masked in grey. The four colored boxes for each season highlight the regions used to extract the time series shown in Figure 14.

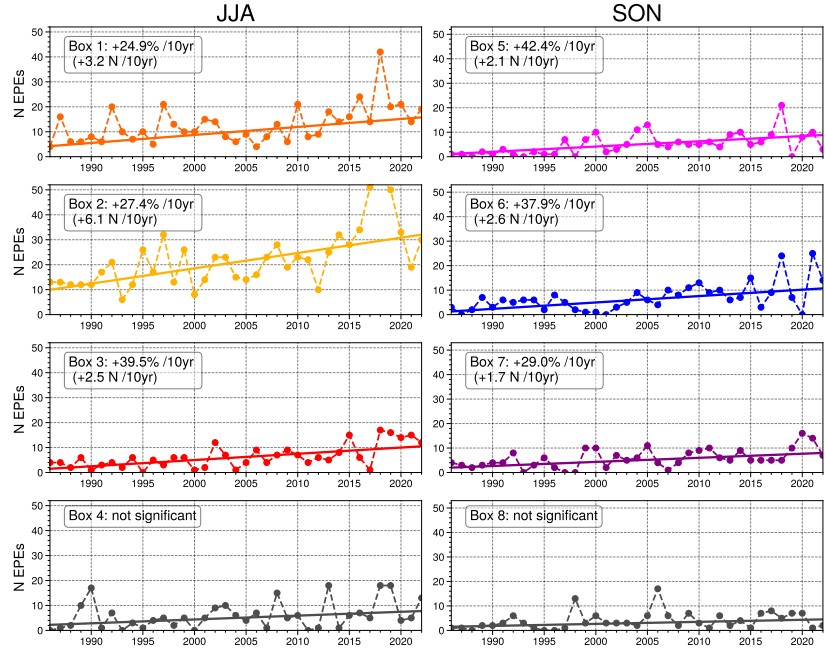

**Figure 14.** Time series of the annual N of EPEs within the colored windows in Figure 13. The left column shows the summer (JJA) series, while the right column displays the autumn (SON) series. Trend lines are plotted for each series, and the corresponding decadal trends are reported. Grey (last row) plots denote non-significant trends.




Apennines, and various coastal and sea areas, such as Ligurian eastern coast, the eastern coast of Sardinia, the southern Adriatic Sea, and the Ionian Sea. Individual series of some selected areas (specifically, inside coloured 0.5 degree windows of Figure 13) are extracted to visualise the EPEs N series along with the detected trends (Figure 14). In summer, trends ranging from 10% to 40%, depending on the region, correspond to an increase of 2 to 6 extreme precipitation events (EPEs) per decade. In

autumn, comparable percentage changes are associated with a smaller increase of 1 to 2 EPEs per decade. In both seasons, some regions also display positive trends that are not statistically significant (e.g., Boxes 4 and 8 in Figure 14), likely due to high interannual variability that dominates the signal. Trends are also computed for the SpS, AvIn, and PkIn of EPEs (see Supplementary material, Figures S8, S9, S10, respectively). Overall, only weak trends (below 10% a decade over Italy) are observed, primarily over land points in summer and over some marine areas in autumn, showing alternating patterns with

a slight tendency toward increasing intensities and decreasing spatial scales. However, none of these trends is statistically significant at any location. This suggests that changes over time are more likely associated with the frequency of EPEs rather than their intensity or spatial extent. It may also reflect the lower noise sensitivity of event counts compared to other indicators. Moreover, trend estimates based on N could be biased by potential double-counting of temporally persistent events, as the analysis is conducted at hourly resolution. To address this, an additional analysis quantifies event persistence, defined as the

number of consecutive hours during which an EPE (i.e., an event exceeding the local average RX1hour threshold) affects the same window.

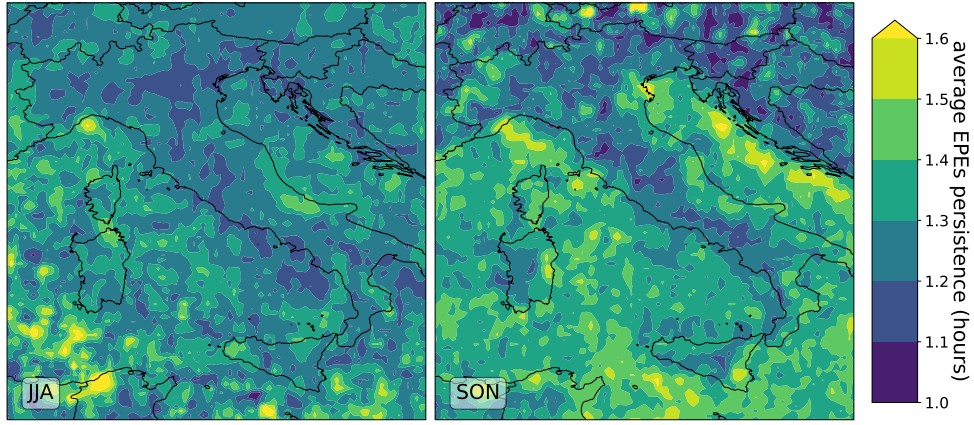

**Figure 15.** seasonal (JJA and SON) maps of the average EPEs persistence (expressed in hours) occurring within the 0.5 × 0.5 window (step size 0.1).

Results (Figure 15) show that persistence exceeds one hour only marginally in most regions, with average persistence values above 1.5 hours limited to a few localised areas expecially during autumn, such as the Ligurian Gulf, where persistent mesoscale convective systems are more common (Cassola et al., 2016), and in parts of eastern Sardinia and southeastern Sicily, where

prolonged convective activity can occur (Forestieri et al., 2018). These findings support the overall temporal isolation of most EPEs and suggest that the impact of double-counting on trend estimates remains limited.





## 4 Discussion

In understanding the results of this work, it is important to underline the uncertainties in analysing signals from the reanalysis representation of hourly precipitation. The MERIDA HRES reanalysis provides hourly precipitation fields over a continuous
and homogeneous 37-year period; however, some limitations affect MERIDA HRES, particularly concerning the representation of precipitation fields at the hourly scale. While the temporal stability and spatial accuracy of MERIDA HRES have been verified in previous studies from climatological to hourly timescales (Cavalleri et al., 2024a; Giordani et al., 2025), it is still necessary to discuss some potential inconsistencies and their impact on the results of this work. Specifically, previous studies have shown that MERIDA HRES systematically overestimates rainfall in summer (and partly in autumn) in regions like the
Po Valley–Adriatic interface, the Calabrian mountains, southern Apulia, and southeastern islands (Cavalleri et al., 2024a; Viterbo et al., 2024; Giordani et al., 2025). These localised wet biases, consistent across timescales, likely stem from overly active explicit convection in the model, as suggested by Figure 8 and 9. This behaviour is sometimes common in WRF-based dynamical downscalings of ERA5 (Bernini et al., 2025), due to some possible problems in representing skin temperature, and difficulty capturing complex land-sea interactions that often arise in an overestimation of precipitation in areas where it is not
normally observed. Even if these aspects need to be taken into account, it is important to notice that these biases are temporally stable and do not coincide spatially with the areas showing significant EPEs increases. Moreover, it is important to underline that ERA5 (Lussana et al., 2024) and its regional downscalings (Cavalleri et al., 2024a) can exhibit stronger precipitation trends than the observed ones. In particular, Cavalleri et al. (2024a) highlighted that the trend in the differences between MERIDA HRES and homogenised observations annual precipitation totals is about $4\%$ for decade, meaning that this fraction of annual
precipitation increase might be attributable to a deviation from observations rather than a true climate signal. This value is not negligible, but overall small if compared to the $10\%$ to $40\%$ increases found in EPEs occurrences. Even with respect to this additional inhomogeneity, the areas affected by it, as delineated in the supplementary material of Cavalleri et al. (2024a), do not overlap with those in which significant trends in EPEs have been found.

The increasing trends in hourly EPEs identified in this study for the period 1986–2022 align with several previous research efforts based on both sub-daily and daily extreme precipitation observations across Italy and its specific regions. In particular, Mazzoglio et al. (2025) reported positive trends in the same Prealpine area analyzed here, based on the RX1hour index, largely attributed to summer convective activity. At a regional scale, Dallan et al. (2022) examined extreme precipitation trends from 1991 to 2020 by separating storm intensity and occurrence frequency, attributing the observed increases in the Eastern Alps
to a growing proportion of sub-daily convective storms during summer. Similarly, Persiano et al. (2020) found a generalized increase in both the frequency and intensity of sub-daily extreme rainfall over the Apennines in Emilia-Romagna (northern Italy) for the 1961–2015 period. Moreover, Pavan et al. (2019), using a daily gridded precipitation dataset for the north and central Italy covering 1961–2015, reported significant positive trends in the 90th percentile of daily precipitation across most of the Alpine area and the northern Po Valley during summer, also supporting the idea that summer and autumn are the seasons



most affected by precipitation changes. In autumn, some of the hourly EPEs trends detected in this study agree with findings by Capozzi et al. (2023), who, based on daily station data for the 2002–2021 period, documented an increasing tendency in both the intensity and frequency of heavy rainfall events in inland Campania. Additionally, the autumnal trends over the central Pre-Alps are in line with the results of Pavan et al. (2019), who also reported significant increases in daily precipitation extremes over the Alps during autumn. This kind of local evidence provides an important observational context that supports

the reliability of some of the signals identified through the present reanalysis-based approach.

## 5   Conclusions

This study employs hourly precipitation fields from the convection-permitting MERIDA-HRES reanalysis to investigate the characteristics of hourly precipitation events, with a focus on their most extreme components and their temporal evolution over the period 1986–2022. This approach yields a twofold outcome. First, it enables the construction of the HOPE-X dataset, an

event-based archive in which nearly 6 million significant precipitation events are described by a set of intensity and spatial characteristics. Second, by isolating the most extreme subset of these events, the method facilitates the description of extreme patterns and the detection of statistically significant trends in the occurrence of EPEs during summer and autumn.

In summer, increasing trends are detected over several Alpine and Prealpine regions as well as in parts of Calabria. In autumn, the most prominent trends emerged over the southern Apennines, over the central Pre-Alps, and several maritime

regions, including Ligurian eastern coast, the eastern coast of Sardinia, the southern Adriatic Sea, and the Ionian Sea.

The results obtained in this work represent an additional perspective within the ongoing and complex debate on precipitation trends in Italy, even with full awareness of some of the limitations of reanalysis datasets. Spatial uncertainty of MERIDA HRES reanalysis was addressed through an event-based approach, which allowed the identification and subsequent spatial aggregation

of hourly events using moving windows, with the intent of reducing the impact of spatial misplacement errors. The results were also interpreted in light of some known and documented local biases of the reanalysis, such as the systematic overestimation of convective precipitation in some areas and some discrepancies between modelled and observed trends at longer timescales. In the final results, regions affected by these model biases were carefully considered in the interpretation of the results and generally did not overlap with the areas where statistically significant trends in the occurrence of EPEs were detected.


The comparison with previous works on precipitation trends and extremes, based on observational data at both daily and sub-daily timescales, supports the robustness of the results presented in this work. In particular, the consistency observed across different studies strengthens the evidence of increasing occurrences of hourly EPEs over specific regions of Italy during summer and autumn.




Future developments may involve leveraging the event-based dataset HOPE-X to explore additional characteristics of EPEs, such as their dominant propagation direction and potential associations with changes in large-scale atmospheric circulation. In selected regions, identifying and employing sufficiently long hourly observational records could allow for a more direct validation of the detected trends. The approach could also be extended to identify EPEs of different nature and duration, including

synoptic-scale events, by analysing longer accumulation periods (e.g., 3, 6, 12, or 24 hours). Furthermore, similar event-based datasets could be produced using the same methodology to detect EPEs in other regions where convection-permitting reanalyses are available.

*Code and data availability.* MERIDA-HRES data are openly available from the RSE repository: "PREC" at https://merida.rse-web.it/. The event-based hourly precipitation dataset used and produced in this study is openly available on Zenodo (DOI: 10.5281/zenodo.15772543), or

via the short link https://bit.ly/HOPE-X. The code used in this work is openly available on GitHub: https://github.com/fcavalleri/EPEs.git.

*Author contributions.* FC: Conceptualization; Formal Analysis; Investigation; Software; Visualization; Writing – Original Draft Preparation; CL: Conceptualization; Methodology; Investigation; Supervision; Validation; Writing – Review & Editing; FV: Data Curation; Methodology; Investigation; Supervision; Validation; Writing – Review & Editing; MB: Investigation; Writing – Review & Editing; RB: Data Curation; Investigation; VM: Investigation; Writing – Review & Editing; ML: Data Curation; Investigation; Funding Acquisition; MM: Investigation;

Supervision; Validation; Writing – Review & Editing; Funding Acquisition.

*Competing interests.* The authors declare that they have no known competing financial interests or personal relationships that could have appeared to influence the work reported in this paper.

*Acknowledgements.* The PhD of the co-author Francesco Cavalleri was activated pursuant to DM 352 and is co-sponsored by PNRR funds and R.S.E. s.p.a. The PNRR funds come from the EU Next Generation programme. This work has been financed by the Research Fund for

the Italian Electrical System under the Three-Year Research Plan 2025-2027 (MASE, Decree n.388 of November 6th, 2024), in compliance with the Decree of April 12th, 2024. This work has been financed by Research Funds from the Italian Ministry for University and Research (PRIN 2022 - CN4RWK – CCHP-ALPS – Climate Change and HydroPower in the Alps, funded by the European Union (Programme Next Generation EU)). Veronica Manara was supported by the "Ministero dell'Università e della Ricerca" of Italy [grant FSE – REACT EU, DM 10/08/2021 n. 1062].

During the preparation of this work, the authors used ChatGPT in order to enhance readability. After using this tool, the authors reviewed and edited the content as needed and take full responsibility for the content of the publication.



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
