# Peer review of "Hourly Precipitation Patterns and Extremization over Italy using convection-permitting reanalysis data"

_EGUsphere, 2025_

## Referee Comment (RC2)

*Review of manuscript egusphere-2025-3455 submitted to Natural Hazards and Earth System Sciences (NHESS)*

**Event-based analysis of extreme precipitation trends in Italy using hourly convection-permitting reanalyses**

by Francesco Cavalleri, et al.

**General comments:**

This manuscript presents an analysis of precipitation event properties (frequency, intensity, duration, and size) using a convection-permitting reanalysis dataset for Italy. The paper is well written, scientifically rigorous, and of clear interest to the NHESS community. That said, the manuscript would benefit from more cautious phrasing regarding the ability of the reanalysis to represent meso-beta scale processes, a more concise and focused introduction, and more consistent use of acronyms. These revisions would improve readability and balance some of the claims. I therefore recommend major revisions before the paper can be considered for publication.

**Specific comment:**

I find the manuscript highly relevant, as it combines two approaches that are still not very common in the community: the high-resolution reanalysis from ERA5 and WRF, and the event-based approach. This framework could be extended to other regions, which increases the value of the study. But the paper would benefit from addressing the following points:

- **Introduction:** The introduction is overly long. Please consider streamlining it to improve readability and focus on the main motivation and novelty of the study.

- **Section 2.1:** Here you present the MERIDA HRES reanalysis, but you do not describe the quality or limitations of the dataset. While some of this is discussed later, it would be more appropriate to include a clear description of dataset strengths and limitations already in this section.

- **Resolution:** The reanalysis resolution is reported as 4 km, presumably referring to the horizontal grid spacing. What about the vertical resolution? A more careful discussion of its role would be valuable. In addition, in the results you claim that the model can resolve meso-beta processes (2–20 km). In my view, a 4 km horizontal resolution may not be sufficient to adequately resolve the full meso-beta range, perhaps only processes above ~10 km. For example, at 4 km resolution, a 10 km feature would be represented by only a handful of grid points. I recommend revising this claim to reflect these limitations.

- **Acronyms:** The manuscript contains many acronyms, which makes it easy to lose track. I recommend reducing their use where possible and writing terms explicitly. I will provide examples in the technical corrections.

**Technical corrections:**

-Line 4: Please specify explicitly how many years of data are available.

-Line 25: Here you state that extreme precipitation changes are due to thermodynamics (CC scaling). What about the role of dynamics? For instance, see Pfahl et al. (2017):

Pfahl, S., O'Gorman, P. A., & Fischer, E. M. (2017). Understanding the regional pattern of projected future changes in extreme precipitation. Nature Climate Change, 7(6), 423-427.

- Line 51: Please, define ARCIS.

- Line 65: Please, define GRIPHO

-Lines 77-79: Please add supporting references.

-Line 147: How sensitive are your results to the choice of the 50th percentile threshold?

-Figure 2: Consider using a single colorbar and enlarging it for clarity.

-Table 1: Consider alternative acronyms instead of axis_maj or lon_wavg, lat_wavg, which are not very intuitive.

-Line 178: Have you considered that grid cell size decreases with latitude? This could affect your area-based results.

-Line 181-182: The acronyms here are difficult to follow, please consider spelling them out.

-Line 230: Please clarify whether this refers to a large quasi-stationary cyclone.

-Line 240: Be cautious with the claim that 4 km simulations can fully resolve these processes; please consider rephrasing.

-Line 374-377: It would be useful to mention dataset biases earlier (e.g. in the data section), and to quantify them explicitly as percentages relative to observations.

-Line 378: Please elaborate on the role of skin temperature in WRF biases.

---

## Referee Comment (RC3)

General comments

The authors make use of a high-resolution reanalysis dataset that represents a considerable effort in terms of length (37 years), spatial coverage (all of Italy), and resolution (4 km). This combination makes it a valuable resource for a national-scale study of changes in extreme precipitation. The availability of hourly rainfall data also makes the study relevant for analyzing the seasonality and spatial variability of hourly precipitation in Italy.

I consider this work useful for the scientific community because it brings together results that were previously limited to local or regional scales into a study covering the whole of Italy. This allows for a better understanding of the spatial variability of extreme precipitation event (EPE) characteristics and their changes, while showing consistency with earlier regional studies.

That said, I find that the Abstract and Introduction could be more explicit regarding the specific scientific contributions and aims of the work. The chosen methodology is interesting in that it introduces a structure-based perspective on hourly precipitation; however, some methodological aspects would benefit from clearer explanation or reconsideration.

Referring to hourly precipitation structures as *"events"* is, in my view, misleading. Since their temporal evolution (duration, displacement, deformation) is not considered, nor the total rainfall volume produced by multi-hour storm systems, the term *"event"* may create confusion and should be replaced with a more precise designation. Finally, the section addressing non-extreme precipitation should be better introduced, with its purpose and relevance clarified.

Overall, in my opinion, the manuscript is suitable for publication after minor revisions. These include adjustments in specific vocabulary, clearer argumentation, additional explanations to help reader understanding, and refinements in the methodology.

Specific comments

1. The 4 km resolution of the reanalysis is at the limit of what is usually called convection-permitting. This should be made explicit in the text, especially since the dataset paper uses "high-resolution" instead. I would keep the term "convection-permitting" for clarity but suggest adding a note that it is at the edge of the definition.

2. Since about half of the figures in the Results section concern hourly rain structures and not EPEs, it is important to either (i) include non-extreme hourly rain discussion in the abstract as well as adapt the title accordingly, or (ii) introduce non-extreme hourly rain results as a necessary step before moving to extremes. Otherwise, the part on extremes, which is announced to the reader, takes too long to arrive.

3. The Introduction and Abstract should state more clearly the purpose of the work and its scientific contribution.

4. The HOPE-X dataset is a collection of hourly rain structures, not "events" in the usual sense of storm systems or local rain events. For instance, a precipitation system lasting two hours can be counted as two "events," even though it is the same system. Similarly, a moving storm can be counted as several "events" in different areas as it displaces. For this reason, the use of the term "events" is inappropriate. The title is also misleading, since the study is not, in my opinion, event-based. The same applies to the dataset name HOPE-X. This does not invalidate the study, but the terminology issue is central and should be revised. I strongly recommend using the term "hourly rain structures." If the term "event" is retained, it should not appear in the title, abstract, or in any part of the text where the definition is not clearly introduced yet (line 164: "Hereafter, the term 'event' denotes the precipitation structures identified using this method"). In that case, it should also be stated that the term "event" is used for readability purposes only.

5. Following the previous comment, in maps such as Figure 6 the number of "events" (N) can reach ~300 per season in a 0.5×0.5 window. This N reflects a mix of the number of hours with a rain structure and the number of surrounding structures at each hour, rather than distinct events. For example, an hour with many small rain structures and an hour with one large rain structure could produce the same hourly rainfall total in that 0.5×0.5 area, yet N would differ greatly. It should therefore either (i) be clarified in the text what the analysis of N is intended to represent if it is something other than the amount of rain, (ii) reconsider the counting methodology, or (iii) include additional analysis showing that N reflects the actual volume of hourly rain.

6. The use of seasonal thresholds to adapt to the intensity scales of each season is well justified and explored. However, the choice to let the seasonal threshold vary spatially with the local 50th percentile of 1 mm+ hourly rains requires stronger justification. As a result of threshold methodology, identical rain structures may be included in HOPE-X in one region but not in another. While such area-relative thresholds are understandable when defining extremes (e.g. EPEs) due to the definition of rarity being potentially region-based, they are harder to justify for

non-extreme rain structures. Choosing fixed threshold across the domain for each season would be more correct. If the authors choose to go with a justification of the current methodological choice of non-homogeneous threshold, consider also that in areas with very little rainfall, the threshold based on wet hours 1mm+ may also be computed on a very small and potentially unrepresentative sample, so this would need to be tested.

7. Potential artifacts due to the threshold methodology chosen discussed in previous point: It is not clear whether some of the patterns in the maps (e.g. the higher JJA values over Ravenna in Fig. 8 AvIn and Fig. 9 PkIn) are genuine results or artifacts of the methodology filtering out structures with higher thresholds (higher thresholds over Ravenna in Fig. 1). Line 257: Are the "hotspots" results of higher local thresholds? Or an actual signal? Same for "isolated spots" line 278.

8. The role of the "minimum enclosing ellipse" is unclear. Its mention in the title of Fig. 2 and in the text (line 169) is confusing: are the variables in Table 1 computed on the ellipse itself or on the rain object it encloses? Is the ellipse only used for visualization of selected structures, or does it play a role in extracting the properties of the rain objects? This part should be clarified for better understanding.

9. Line 169: for readers unfamiliar with Wernli et al. (2008), it would be helpful if the purpose of using this methodology were explained.

10. Following the previous points, it is unclear whether *axis_maj* refers to the major axis of (i) the rain structure itself or (ii) the enclosing ellipse. In both cases, the phrasing in line 264 ("summer events are generally smaller") is problematic, as are later formulations in the SpS analysis regarding size. If (i), and the aim is to characterize the size of the structures, the "area" variable would be more appropriate. For example, a thin elongated structure may be characterized as large with SpS even though its area of coverage is small, while a rounder structure with the same SpS value could cover a much larger area, yet still be classified as the same "size" when using SpS. If (ii), the analysis would instead describe the size of influence of the structure rather than its actual size. It should therefore be clarified what SpS is intended to capture, and statements about "size" should be made with caution. Replacing the SpS analysis with an analysis of the area variable would make the interpretation more straightforward.

Technical comments

1. Line 10: "the most extreme component." The word *component* is unclear here. I suggest rephrasing to: *"The most extreme rain events (EPEs)…"*

2. Line 111: "event-based approach." As discussed earlier, this would be better described as an approach using hourly rain spatial structures. I also suggest adapting the phrase to be less generic and more explicit about how clustering helps address the limitations:
   *"In light of these limitations, an object-based approach using a clustering technique was adopted to capture coherent hourly precipitation structures and reduce sensitivity to small-scale discrepancies between simulations and observations."*

3. Line 146: "The event-detection … precipitation events…" The vocabulary of *event* throughout the text before line 164 is misleading, since line 164 specifies:
   *"Hereafter, the term 'event' denotes the precipitation structures identified using this method."* Event terminology should not appear earlier without this clarification.

4. Line 296: "is conducted on a subset of EPEs." A clearer phrasing would be: *"a subset of the HOPE-X dataset."* (But HOPE-X should be renamed)

5. Lines 321–322: The phrase *"seasonal redistribution is likely driven by the persistence of summer-like convective activity into early autumn"* is confusing, since the focus is on differences between summer and autumn. This cannot be justified by describing autumn activity as "summer-like." A phenomenon-based explanation could instead highlight that the persistence of warm sea surface conditions beneath a cooler atmosphere creates instability favorable to convection. Autumnal convection is not "summer-like". This is also reflected in Fig. 12, which shows differences in the "size" characteristics of hourly rain structures between summer and autumn.

6. Line 354: Instead of *"alternating pattern,"* I suggest: *"spatial heterogeneity in the sign of the signal"* or *"patterns of alternating signs."*

---

## Author Comment (AC1)

**Review of manuscript egusphere-2025-3455 submitted to Natural Hazards and Earth System Sciences (NHESS)**

**Anonymous Referee #1, 28 Aug 2025**

[answers in blue]

The manuscript presents reanalyses model simulation results on convective-permitting scales over Italy with hourly resolution. The study is important since understanding precipitation and provides a method of studying events at hourly scales. Extreme precipitation trends on an hourly basis are important for assessing flood risk, especially as these trends are expected to increase with a warming climate, even in drying areas. The paper is generally well written, and the figures are well presented, however there are a few major concerns, along with some minor comments, that I hope the authors will address in order for the article to be suitable for publication.

**General comments**

A major concern is that the title and abstract of the paper describe extremes, while a lot of the results and figures describes the full dataset compared to the extreme analysis. The Result section should be altered to highlight the extreme analyses better and maybe reduce the description and figures around the full dataset to better suit the journal. Another major concern is that with the weight given on extremes in titles and abstract, there is only one threshold for extremes used which also is set very high, especially when considering hourly data. Above the mean of 37 datapoints which give only 18-19 events per grid cell if the events are evenly distributed around the mean (a quick calculation sets this threshold around the 9.995 percentile). The threshold applied to define extreme precipitation is exceptionally high, and with so few events included, the resulting trend estimates are highly uncertain. The robustness of the study would be improved if additional analyses were carried out using several lower thresholds, allowing for a more comprehensive assessment of trends.

We would like to sincerely thank the reviewer for the careful and thorough reading of our manuscript and for the constructive comments, which will greatly help us improve the quality of the paper.

Regarding the first major concern, we acknowledge that a considerable part of the Results section is devoted to the analysis of the full event-based dataset. Our intention in doing so was to show that the identified precipitation events are consistent with the climatological characteristics of hourly precipitation over Italy, and therefore to provide a solid basis for the subsequent analysis of extremes. Moreover, since we shared this dataset with the scientific community, a complete explanation of it is necessary. Nevertheless, we agree with the reviewer that, given the emphasis on extremes in the title and abstract, the Result section of the manuscript can be better aligned with this aim. In the revised version, we will reduce the number of figures and the descriptive content related to the full dataset, potentially moving some material to the Supplementary Material, in order to dedicate more space and emphasis to the analyses of the subset of events classified as extremes.

Regarding the second major concern, we understand the reviewer's point about the use of a high threshold for defining extremes. It is correct that our threshold corresponds to approximately 18–19 events per grid cell if events were evenly distributed around the mean. However, we may not have emphasized enough in the manuscript that extremes are not selected point by point. Instead, we follow an event-based approach, where an event is defined as a cluster of many points and is classified as an Extreme Precipitation Event (EPE) if at least one grid point within the cluster (the one with the peak value) exceeds the threshold. Combined with the fact that peak value distributions are typically strongly right-skewed with fat tails (see Coles, 2001), this leads to a

selection of EPEs that is less restrictive than it may initially seem. For instance, about 11% of summer events in the full dataset exceeds the average RX1hour threshold. This reflects the choice of a fixed threshold applied to events that vary seasonally: fewer events occur in summer, but they tend to be more intense (see Results, Section 3.1).

We acknowledge that lowering the threshold would increase the number of identified extremes, but we believe this would blur the distinction between EPEs, as described by the Extreme Events Theory (see Coles, 2001), and more moderate high-quantile events, thereby reducing interpretability. Our decision to adopt a fixed threshold based on the mean of annual maxima (RX1hour) was also guided by the approach proposed by Lavers et al. (2025) at the European scale, where fixed thresholds (in that case RX1day) were explicitly recommended for their clarity in communicating extremes.

We also recognize that the relatively small sample size of extremes may raise concerns about statistical robustness. To address this, and to mitigate spatial uncertainty in reanalysis data, we applied a moving window spatial aggregation (0.5° × 0.5°) and employed state-of-the-art trend detection methods: Sen's non-parametric slope estimate, Mann–Kendall significance testing, and field significance analysis with False Discovery Rate control (Wilks, 2006). This multi-method framework supports the statistical robustness of our trend results despite sample limitations.

For these reasons, we believe that our choice of a fixed, high threshold ensures both conceptual rigor and clarity in the identification of extreme precipitation events. We will provide better explanation of our choices regarding this aspect in the revised manuscript.

Coles, S. (2001), An Introduction to Statistical Modeling of Extreme Values, Springer-Verlag, London, https://link.springer.com/book/10.1007/978-1-4471-3675-0

Lavers, D. A., Villarini, G., Cloke, H. L., Simmons, A., Roberts, N., Lombardi, A., Burgess, S. N., & Pappenberger, F. (2025). How bad is the rain? Applying the extreme rain multiplier globally and for climate monitoring activities. Meteorological Applications, 32(2), e70031. https://doi.org/10.1002/met.70031

Sen, P. K.(1968), Estimates of the Regression Coefficient Based on Kendall's Tau, Journal of the American Statistical Association, 63, 1379–1389, https://doi.org/10.1080/01621459.1968.10480934

McLeod, A. I.: Kendall rank correlation and Mann-Kendall trend test, R package Kendall, 602, 1–10, 2005.

Wilks, D. S., 2006: On "Field Significance" and the False Discovery Rate. J. Appl. Meteor. Climatol., 45, 1181–1189, https://doi.org/10.1175/JAM2404.1

**Specific Comments**

Page 2. L27-29: Even drying areas experience more extreme precipitation events.

We thank the reviewer for this useful remark. We will revise the sentence accordingly to explicitly acknowledge that even regions undergoing drying trends may experience an intensification of extreme precipitation events, as noted e.g. by Donat et al. (2016)

Donat, M., Lowry, A., Alexander, L., O'Gorman, P., & Maher, N. (2016). More extreme precipitation in the world's dry and wet regions. *Nature Climate Change*, 6, 508-513. https://doi.org/10.1038/NCLIMATE2941

Page 5. Section 2.2  Median is easier than the 50$^{th}$ percentile, and used earlier in Introduction.

We appreciate the reviewer's suggestion. We agree that the use of the term median is more appropriate and clearer than 50th percentile. In the revised version, we will therefore adopt the term "median" throughout the manuscript, while keeping the expression 50th percentile only at its first occurrence to clarify the statistical meaning.

The Method section needs to be improved to better understand the results presented. Section 2.2 should be rewritten to increase readability. The thresholds and smoothing are first presented, and then described again in the later paragraph, maybe rewrite for better readability.

Concerning the organization of Section 2.2, we fully acknowledge the need for improved readability. Following the reviewer's advice, we will restructure the section so that the thresholds and their smoothing are introduced and described in the same place, thereby avoiding repetition and facilitating a more coherent presentation.

Line 162 – 165 More specifically, … this sentence in especially hard to follow. More than half in which instances?

We thank the reviewer for pointing out the lack of clarity in this sentence. What we intended to convey is that an event is considered *relevant* when the precipitation exceeds the median of the distribution of events greater than 1 mm within a given area and season. We agree that the current formulation is difficult to follow, and we will therefore rephrase the sentence in the revised manuscript to ensure the definition is stated more clearly and unambiguously.

I'm also curious how time is handled as an event usually lasts more than one hour. Is there any clustering in time?

We thank the reviewer for raising this important point. In our analysis, the general approach was to study events at the hourly scale, regardless of what happens before or after. However, the aspect of event duration is explicitly addressed in the final part of Section 3.3, where we analyze the average persistence of extreme events (Figure 15). This analysis provides an estimate of the mean duration of the extremes as defined in our study and shows that such events typically last only slightly longer than one hour on average. This finding not only clarifies the temporal scale of our definition of extremes but also helps ensure the robustness of the calculated trends by reducing the risk of double counting. In the revised version, we will emphasize both the strengths and limitations of this choice and note that future work could benefit from explicitly applying temporal clustering techniques.

Section 2.4 Is this Gaussian filter the same as used in section 2.2? If so, does the justification of this radius apply to the earlier smoothing, and move this part in section 2.2?

We thank the reviewer for this observation. Yes, the Gaussian filter applied in Section 2.4 is the same as the one used in Section 2.2, and the justification of the chosen radius is identical in both cases. To avoid redundancy and improve clarity, in the revised version we will unify the explanation and provide the motivation in a single place, making explicit reference to it in both sections where the filter is applied.

Table 2, The table could be improved if the names were included in addition to the short names.

We thank the reviewer for this helpful suggestion. In the revised manuscript, we will add an additional column including the full names alongside the short names to improve the readability of the table.

Page 9. L196 ERA5 is already introduced as the driver of the dataset.

We thank the reviewer for this useful remark. In the revised manuscript, we will remove the phrase *"which is a global reanalysis product with a 0.25° grid spacing"* to avoid repeating information already introduced earlier when describing the ERA5 dataset.

Page 10, l207-208 Explain better here: How many events would this give over a typical grid cell on the coast or in the mountains?

We appreciate the reviewer's request for clarification. As already discussed in response to the second major concern, extremes are not selected point by point. Instead, we adopt an event-based approach, in which an event is defined as a cluster of grid points and is classified as an Extreme Precipitation Event (EPE) if at least its peak exceeds the threshold. Moreover, precipitation distributions are typically right-skewed with heavy tails and vary both seasonally and regionally. This leads to a higher number of extremes (compared to the 17–18 events expected from point-by-point selection under symmetric distributions) and to substantial variability across regions (e.g., coastal vs. mountainous areas). This variability is evident in Figure 11. In this analysis the number of occurrences is not computed at the single grid-cell level, but within a 0.5° moving window (≈156 grid points) in order to reduce the impact of misplacement errors. In the revised manuscript, we will make this clearer at the end of Section 2.4 to improve readability and ensure that the methodology is transparent to the reader.

Page 20. L 358-359: This analysis would also benefit from a lower threshold, as two executive hours above RX1hour is extremely rare.

We agree with the reviewer that two consecutive hours above the average RX1hour are indeed extremely rare. However, this rarity is consistent with our methodological choice: by construction, extremes are derived from the mean of hourly annual maxima (RX1hour) and are meant to identify only the most exceptional events, in line with Extreme Value Theory. This makes the occurrence of consecutive exceedances intrinsically unlikely, but at the same time confirms that the analysis truly focuses on *extremes*, and supports the approximation of considering extremes of different hours as independent.

As discussed in our response to the second major concern, lowering the threshold would increase the number of identified extreme events but would also blur the distinction between genuine extremes and more moderate high-quantile precipitation events. From a practical perspective, it would also bring to more multi-hour episodes, requiring the adoption of event-tracking techniques, which falls outside the scope of this work. For these reasons, we consider our current approach the most appropriate for capturing the statistical behavior of precipitation extremes at the hourly scale.

Page 22. L 423-424, This sentence could be misinterpreted, there could be trends here that were not found because of issues with the reanalysis. Could there be a false positive trend, or a false insignificant trend?

We thank the reviewer for this important observation. We agree that the limitations of reanalysis data may indeed have masked existing trends, leading to the possibility of *false insignificant trends*, i.e. trends that are in fact present and significant but cannot be robustly detected due to the constraints of the dataset. What we intended to stress in that line is that the significant trends we do identify remain robust even when accounting for the specific limitations of the reanalysis. To avoid misinterpretation, we will reformulate the sentence in the revised manuscript to better convey this distinction.

---

## Author Comment (AC2)

**Review of manuscript egusphere-2025-3455 submitted to Natural Hazards and Earth System Sciences (NHESS)**

**Anonymous Referee #2, 1 Sep 2025**

[answers in blue]

**General comments:** This manuscript presents an analysis of precipitation event properties (frequency, intensity, duration, and size) using a convection-permitting reanalysis dataset for Italy. The paper is well written, scientifically rigorous, and of clear interest to the NHESS community. That said, the manuscript would benefit from more cautious phrasing regarding the ability of the reanalysis to represent meso-beta scale processes, a more concise and focused introduction, and more consistent use of acronyms. These revisions would improve readability and balance some of the claims. I therefore recommend major revisions before the paper can be considered for publication.

**Specific comment**: I find the manuscript highly relevant, as it combines two approaches that are still not very common in the community: the high-resolution reanalysis from ERA5 and WRF, and the event-based approach. This framework could be extended to other regions, which increases the value of the study. But the paper would benefit from addressing the following points:

- Introduction: The introduction is overly long. Please consider streamlining it to improve readability and focus on the main motivation and novelty of the study.

Thank you for your constructive and insightful comments. We appreciate your recognition of the relevance and potential of our study, as well as your detailed suggestions for improvement. We will carefully revise the manuscript to implement your recommendations, in particular by streamlining and focusing the introduction.

- Section 2.1: Here you present the MERIDA HRES reanalysis, but you do not describe the quality or limitations of the dataset. While some of this is discussed later, it would be more appropriate to include a clear description of dataset strengths and limitations already in this section.

We agree that Section 2.1 would benefit from a clearer description of the strengths and limitations of the MERIDA HRES reanalysis. While these aspects have already been extensively discussed in our previous works (Cavalleri et al., 2024; Viterbo et al., 2024), in the revised version of the manuscript we will expand this section to include a concise overview of the dataset's quality—highlighting both its ability to capture precipitation processes at convection-permitting scales and its known limitations. We believe that relocating part of this information (currently presented in the Discussion) to an earlier section will improve clarity and allow readers to better evaluate the results.

Cavalleri, F., C. Lussana, F. Viterbo, M. Brunetti, R. Bonanno, V. Manara, M. Lacavalla, S. Sperati, and M. Raffa (2024). Multi-scale assessment of high-resolution reanalysis precipitation fields over Italy. Atmospheric Research, 312, 107734. https://doi.org/10.1016/j.atmosres.2024.107734

Viterbo, F., S. Sperati, B. Vitali, F. D'Amico, F. Cavalleri, R. Bonanno, and M. Lacavalla (2024). MERIDA HRES: A New High-Resolution Reanalysis Dataset for Italy. Meteorological Applications, 31(6), e70011. https://doi.org/10.1002/met.70011

- Resolution: The reanalysis resolution is reported as 4 km, presumably referring to the horizontal grid spacing. What about the vertical resolution? A more careful discussion of its role would be valuable. In addition, in the results you claim that the model can resolve meso-beta processes (2–20 km). In my view, a 4 km horizontal resolution may not be sufficient to adequately resolve the full

meso-beta range, perhaps only processes above ~10 km. For example, at 4 km resolution, a 10 km feature would be represented by only a handful of grid points. I recommend revising this claim to reflect these limitations.

Thank you for this important observation. We agree that additional detail on the resolution of MERIDA HRES is needed. In the revised manuscript, we will explicitly state that MERIDA HRES has 56 vertical levels, with increased vertical resolution in the lower atmosphere (e.g., levels located at 10, 35, 70, 100, 130, 180, 250, 325, 415, and 500 m; Viterbo et al., 2024). The effective horizontal spatial resolution of MERIDA HRES has been thoroughly evaluated in one of our previous work using a wavelet spectral decomposition approach (Cavalleri et al., 2024b), which demonstrated the dataset's ability to represent convective precipitation events. At the same time, we will revise our discussion of meso-beta processes, expanding this discussion in the revised version of the manuscript to provide a more complete and balanced assessment of the dataset's capabilities and limitations.

Viterbo, F., S. Sperati, B. Vitali, F. D'Amico, F. Cavalleri, R. Bonanno, and M. Lacavalla (2024). *MERIDA HRES: A New High-Resolution Reanalysis Dataset for Italy*. Meteorological Applications, 31(6), e70011. https://doi.org/10.1002/met.70011

- Acronyms: The manuscript contains many acronyms, which makes it easy to lose track. I recommend reducing their use where possible and writing terms explicitly. I will provide examples in the technical corrections.

Thank you for this helpful suggestion. Reducing the number of acronyms and writing terms explicitly where possible will improve the readability of the manuscript. In the revised version, we will carefully review the text and limit the use of acronyms, while ensuring that the key terms remain clear and consistent throughout the paper.

**Technical corrections:**

-Line 4: Please specify explicitly how many years of data are available.

Thank you for pointing this out. In the revised manuscript, we will explicitly indicate that the dataset spans 37 years, from 1986 to 2022.

-Line 25: Here you state that extreme precipitation changes are due to thermodynamics (CC scaling). What about the role of dynamics? For instance, see Pfahl et al. (2017):

Pfahl, S., O'Gorman, P. A., & Fischer, E. M. (2017). Understanding the regional pattern of projected future changes in extreme precipitation. Nature Climate Change, 7(6), 423-427.

Thank you for this important comment and for suggesting a valuable reference. We agree that extreme precipitation changes are influenced not only by thermodynamic effects, such as Clausius–Clapeyron scaling, but also by dynamic processes. While thermodynamic effects tend to produce a relatively uniform fractional increase in extremes, dynamic contributions can modify these changes regionally, leading to variations in the intensity and frequency of extreme events. In the revised manuscript, we will clarify this point and discuss both thermodynamic and dynamic influences on extreme precipitation.

- Line 51: Please, define ARCIS.

Thank you for pointing this out. In the revised manuscript, we will explicitly define ARCIS as the "*Archivio Climatologico per l'Italia Centro Settentrionale*", a high-resolution climate precipitation dataset for north-central Italy (Pavan et al., 2019)

Pavan, V., Antolini, G., Barbiero, R., Berni, N., Brunier, F., C., A. Cagnati, O. Cazzuli, A. Cicogna, C. De Luigi, E. Di Carlo, M. Francioni, L. Maraldo, G. Marigo, S. Micheletti, L. Onorato, E. Panettieri, U. Pellegrini, R. Pelosini, D. Piccinini, S. Ratto, C. Ronchi, L. Rusca, S. Sofia, M. Stelluti, R. Tomozeiu, T. Torrigiani Malaspina, 2019: *High resolution climate precipitation analysis for north-central Italy, 1961–2015.* Clim. Dyn. 52, 3435–3453. https://doi.org/10.1007/s00382-018-4337-6

- Line 65: Please, define GRIPHO

In the revised manuscript, we will explicitly define GRIPHO as "GRidded Italian Precipitation Hourly Observations", a high-resolution precipitation dataset for Italy (Fantini, 2019).

Fantini, A. *Climate change impact on flood hazard over Italy*, PhD Thesis, (2019).

-Lines 77-79: Please add supporting references.

Thank you for this suggestion. We agree that providing supporting references will strengthen these statements. In the revised manuscript, we explain that operative rain gauge networks tend to underestimate extreme convective precipitation over small areas, and point-scale extremes can be underestimated by about 20% (Schroeer et al., 2018). Conversely, radar and satellite-based precipitation estimates may exhibit large positive or negative biases (Wang et al., 2021).

Schroeer, K., Kirchengast, G., & O, S. (2018). *Strong Dependence of Extreme Convective Precipitation Intensities on Gauge Network Density.* Geophysical Research Letters, 45, 8253–8263. https://doi.org/10.1029/2018GL077994

Wang, S., Li, C., Li, D., Tian, X., Bao, H., Chen, G., & Xia, Y. (2021). *Exploring the utility of radar and satellite-sensed precipitation and their dynamic bias correction for integrated prediction of flood and landslide hazards.* Journal of Hydrology. https://doi.org/10.1016/j.jhydrol.2021.126964

-Line 147: How sensitive are your results to the choice of the 50th percentile threshold?

Thank you for this comment. We have tested the sensitivity of our results building alternative event-based datasets constructed with different thresholds, both percentile-based and fixed. In particular, as noted in lines 286–290 of the manuscript, using a fixed 1 mm threshold (figures provided in the Supplementary Material), our analyses show small variations, indicating that the observed seasonal differences primarily reflect genuine variability rather than artifacts introduced by the clustering method. Importantly, this also holds for extreme precipitation trends, suggesting that our results are robust to the choice of the 50th percentile threshold.

-Figure 2: Consider using a single colorbar and enlarging it for clarity.

Thank you for this suggestion. In the revised manuscript, we will adjust Figure 2 accordingly to enhance readability.

-Table 1: Consider alternative acronyms instead of axis_maj or lon_wavg, lat_wavg, which are not very intuitive.

Thank you for the suggestion. The acronyms $axis\_maj$, $lon\_wavg$, and $lat\_wavg$ are those used in the HOPE-X dataset, and we need to retain them to maintain consistency and facilitate reference to the original data. In the rest of the text, we will strive to use more descriptive terms to ensure clarity.

-Line 178: Have you considered that grid cell size decreases with latitude? This could affect your area-based results.

Thank you for the comment. We have considered the variation of grid cell size with latitude. Across our study area (36°N–47°N), the relative difference in grid cell area due to meridian convergence is limited, with cells at higher latitudes being slightly smaller. To quantify this effect, we used the NOAA distance calculator (https://www.nhc.noaa.gov/gccalc.shtml) and verified that the distance represented by 0.1° in longitude decreases only modestly with latitude: for example, from approximately 9 km at 36°N to about 8 km at 47°N.

For many large-scale analyses, this difference (on the order of ~10–15%) is considered negligible, particularly when results are aggregated or averaged across the domain. Our objective is to provide a consistent view of the relationship between spatial scales and the characteristics of the identified events. In this context, the modest variations in grid cell shape and size do not significantly affect our conclusions.

-Line 181-182: The acronyms here are difficult to follow, please consider spelling them out.

Thank you for the comment. We will follow your suggestion and avoid using acronyms here, spelling them out instead to improve clarity, as also suggested by reviewer #1.

-Line 230: Please clarify whether this refers to a large quasi-stationary cyclone.

Thank you for this comment. You are correct: it refers to a large quasi-stationary cyclone. We will clarify this point in the revised manuscript to make it explicit.

-Line 240: Be cautious with the claim that 4 km simulations can fully resolve these processes; please consider rephrasing.

Thank you for the comment. We agree that the original statement may sound too strong. In the revised manuscript, we will rephrase it more cautiously to reflect that 4 km simulations are able to *explicitly represent* convection and related processes, but *may not fully resolve* them. This wording better captures the current understanding of the capabilities and limitations of convection-permitting simulations.

-Line 374-377: It would be useful to mention dataset biases earlier (e.g. in the data section), and to quantify them explicitly as percentages relative to observations.

Thank you for the comment. We agree that discussing dataset biases earlier in the manuscript would improve clarity. In the revised version, we will introduce this information already in the Data section and provide a quantification of the biases as percentages relative to observations.

-Line 378: Please elaborate on the role of skin temperature in WRF biases.

Thank you for the comment. In the revised manuscript, we will elaborate on the role of skin temperature in WRF biases, highlighting how deviations in surface temperature can propagate to near-surface variables and influence model performance.

---

## Author Comment (AC4)

**Review of manuscript egusphere-2025-3455 submitted to Natural Hazards and Earth System Sciences (NHESS)**

**Anonymous Referee #3, 19 Sep 2025**

[answers in blue]

**General comments**

The authors make use of a high-resolution reanalysis dataset that represents a considerable effort in terms of length (37 years), spatial coverage (all of Italy), and resolution (4 km). This combination makes it a valuable resource for a national-scale study of changes in extreme precipitation. The availability of hourly rainfall data also makes the study relevant for analyzing the seasonality and spatial variability of hourly precipitation in Italy.

I consider this work useful for the scientific community because it brings together results that were previously limited to local or regional scales into a study covering the whole of Italy. This allows for a better understanding of the spatial variability of extreme precipitation event (EPE) characteristics and their changes, while showing consistency with earlier regional studies.

That said, I find that the Abstract and Introduction could be more explicit regarding the specific scientific contributions and aims of the work. The chosen methodology is interesting in that it introduces a structure-based perspective on hourly precipitation; however, some methodological aspects would benefit from clearer explanation or reconsideration.

Referring to hourly precipitation structures as "events" is, in my view, misleading. Since their temporal evolution (duration, displacement, deformation) is not considered, nor the total rainfall volume produced by multi-hour storm systems, the term "event" may create confusion and should be replaced with a more precise designation. Finally, the section addressing non-extreme precipitation should be better introduced, with its purpose and relevance clarified.

Overall, in my opinion, the manuscript is suitable for publication after minor revisions. These include adjustments in specific vocabulary, clearer argumentation, additional explanations to help reader understanding, and refinements in the methodology.

**Specific comments**

The 4 km resolution of the reanalysis is at the limit of what is usually called convection-permitting. This should be made explicit in the text, especially since the dataset paper uses "high-resolution" instead. I would keep the term "convection-permitting" for clarity but suggest adding a note that it is at the edge of the definition.

Thank you for this helpful suggestion. In the revised manuscript, we will adopt the term *convection-permitting* instead of *high-resolution* when referring to MERIDA HRES. As also discussed in our response to Reviewer 2, we acknowledge that the 4 km horizontal resolution lies at the boundary of the convection-permitting definition, and we will make this explicit in the text. This clarification will ensure consistency and transparency regarding the dataset's capabilities and limitations (Viterbo et al., 2024; Cavalleri et al., 2024b).

Cavalleri, F., C. Lussana, F. Viterbo, M. Brunetti, R. Bonanno, V. Manara, M. Lacavalla, S. Sperati, and M. Raffa (2024). *Multi-scale assessment of high-resolution reanalysis precipitation fields over Italy*. Atmospheric Research, 312, 107734. https://doi.org/10.1016/j.atmosres.2024.107734

Viterbo, F., S. Sperati, B. Vitali, F. D'Amico, F. Cavalleri, R. Bonanno, and M. Lacavalla (2024). *MERIDA HRES: A New High-Resolution Reanalysis Dataset for Italy*. Meteorological Applications, 31(6), e70011. https://doi.org/10.1002/met.70011

Since about half of the figures in the Results section concern hourly rain structures and not EPEs, it is important to either (i) include non-extreme hourly rain discussion in the abstract as well as adapt the title accordingly, or (ii) introduce non-extreme hourly rain results as a necessary step before moving to extremes. Otherwise, the part on extremes, which is announced to the reader, takes too long to arrive.

We will include in the Abstract a mention of the findings related to non-extreme hourly precipitation structures (i) and, also following Reviewer 1 suggestion, we will make the motivation for including non-extreme rainfall analysis clearer (ii) at the beginning of that section, while at the same time shortening its discussion so that the focus on extremes—highlighted in the title and Abstract—emerges earlier in the Results section.

The Introduction and Abstract should state more clearly the purpose of the work and its scientific contribution.

In the revised manuscript, we will add a specific sentence in the Abstract to clarify the purpose of the study, namely to contribute to the discussion on extreme precipitation trends in Italy through an innovative approach based on hourly fields from a convection-permitting reanalysis. In addition, we will expand the relevant paragraph in the Introduction to more explicitly state the aim and scientific contribution of the work.

The HOPE-X dataset is a collection of hourly rain structures, not "events" in the usual sense of storm systems or local rain events. For instance, a precipitation system lasting two hours can be counted as two "events," even though it is the same system. Similarly, a moving

storm can be counted as several "events" in different areas as it displaces. For this reason, the use of the term "events" is inappropriate. The title is also misleading, since the study is not, in my opinion, event-based. The same applies to the dataset name HOPE-X. This does not invalidate the study, but the terminology issue is central and should be revised. I strongly recommend using the term "hourly rain structures." If the term "event" is retained, it should not appear in the title, abstract, or in any part of the text where the definition is not clearly introduced yet (line 164: "Hereafter, the term 'event' denotes the precipitation structures identified using this method"). In that case, it should also be stated that the term "event" is used for readability purposes only.

We thank the Reviewer for this valuable comment and agree that a more precise terminology is appropriate for describing the dataset of all identified structures. In the revised manuscript, we will therefore refer to them as *Hourly Precipitation Spatial Structures (HPSSs)* instead of "events," and use *Hourly Precipitation Extremes (HPEs)* rather than "extreme precipitation events (EPEs)."

Nevertheless, we would like to retain a reference to "extreme events" in specific contexts, as this terminology is widely used in the literature and reflects different methodological approaches to defining rainfall events. For instance, some studies follow the motion of precipitation systems or clusters in a Lagrangian framework, whereas others adopt a fixed-area perspective, leading to a more local Eulerian definition (e.g., Ignaccolo & De Michele, 2010). Our study follows the latter approach: we do not track the temporal evolution of precipitation systems nor attempt to follow individual storms as they move. Instead, we adopt the viewpoint of a fixed spatial reference (i.e. the averaging window) and focus on the spatial structures occurring at that location.

When focusing on extremes, the HPSSs exceeding the extreme threshold (and so defined as HPEs) are typically short-lived: it is rare for the threshold to be exceeded locally for more than one hour. This is already partly illustrated in the mean persistence analysis (Fig. 15). To further support this point, we now also provide the distribution of HPE durations within 0.1° windows for JJA and SON:

[Figure]

[Figure]

This underscores that HPEs almost always last only one hour and can therefore still be meaningfully associated with the concept of an "extreme event" in an Eulerian framework.

In the revised manuscript, we will (i) consistently replace the term "event" with Hourly Precipitation Spatial Structure (HPSS) when referring to the full dataset, (ii) use Hourly Precipitation Extremes (HPE) instead of "extreme precipitation events (EPEs)," and (iii) clarify the methodological framework and definition of "extreme event" in the Introduction and Methods sections, not using the word "event" before this explanation (not in the title nor in the abstract).

Ignaccolo, M. & De Michele, C., 2010. A point based Eulerian definition of rain event based on statistical properties of inter drop time intervals: An application to Chilbolton data. *Advances in Water Resources*, 33(8), pp.933–941. doi:10.1016/j.advwatres.2010.04.002

Following the previous comment, in maps such as Figure 6 the number of "events" (N) can reach ~300 per season in a 0.5×0.5 window. This N reflects a mix of the number of hours with a rain structure and the number of surrounding structures at each hour, rather than distinct events. For example, an hour with many small rain structures and an hour with one large rain structure could produce the same hourly rainfall total in that 0.5×0.5 area, yet N would differ greatly. It should therefore either (i) be clarified in the text what the analysis of N is intended to represent if it is something other than the amount of rain, (ii) reconsider the counting methodology, or (iii) include additional analysis showing that N reflects the actual volume of hourly rain.

Thank you for raising this important point. The purpose of this analysis is to identify the locations where rainfall structures occur most frequently, by counting the number of centers of mass falling within each grid window. This metric is therefore distinct from total rainfall volume, which is investigated through the mean intensity analysis. To explore alternative approaches, we also tested the frequency of wet hours (i.e., the percentage of hours in which at least one structure was detected within the window), which helps disentangle the mix you mention in your comment. The resulting spatial patterns were very similar to those obtained in Figure 6:

[Figure]

For the sake of simplicity, and to keep the analysis more directly aligned with the increase in the number of extreme events—which we find conceptually more tangible than frequency—we chose to present results in terms of the number of events (N) rather than wet-hour frequency. This point will be clarified explicitly when introducing Figure 6 in the revised manuscript.

The use of seasonal thresholds to adapt to the intensity scales of each season is well justified and explored. However, the choice to let the seasonal threshold vary spatially with the local 50th percentile of 1 mm+ hourly rains requires stronger justification. As a result of threshold methodology, identical rain structures may be included in HOPE-X in one region but not in another. While such area-relative thresholds are understandable when defining extremes (e.g. EPEs) due to the definition of rarity being potentially region-based, they are harder to justify for non-extreme rain structures. Choosing fixed threshold across the domain for each season would be more correct. If the authors choose to go with a justification of the current methodological choice of non-homogeneous threshold, consider also that in areas with very little rainfall, the threshold based on wet hours 1mm+ may also be computed on a very small and potentially unrepresentative sample, so this would need to be tested.

Thank you for this detailed and thoughtful comment. The choice of threshold for identifying coherent precipitation structures was extensively discussed among the co-authors, reflecting the diversity of perspectives within our team. The final decision followed a series of sensitivity tests, some of which are included in the Supplementary Material. These tests compared fixed thresholds (0.5, 1, and 2 mm) with percentile-based thresholds (50th, 75th, 90th, and 99th percentiles), computed both from all values and from values above 1 mm.

We opted for spatially varying thresholds for the same rationale as for seasonal thresholds. Precipitation regimes in Italy differ markedly across both seasons and regions, due to the country's complex geography (latitudinal gradients, mountain ranges, coastlines, inland areas). Even within the same season, distinct precipitation processes coexist in different regions. A threshold that varies seasonally and spatially therefore ensures that the identified

structures are meaningful for the local climatology and meteorology. The aim of this step is to retain precipitation objects of sufficient relevance for the season and region in question. Spatial modulation also improves the identification of coherent structures and helps separate nearby systems—for example, distinguishing individual thunderstorms within a mesoscale cluster in areas where median precipitation is higher—thus avoiding their artificial merging into a single object. These aspects will be clarified in the revised manuscript.

Regarding regions with very little rainfall (e.g., over the sea in summer), we acknowledge the reviewer's concern. Our full-dataset analysis (Section 3.1) indeed shows a scarcity of identified structures in such areas. When extreme events are subsequently filtered, these regions are excluded from the trend calculations (grey areas in Fig. 13), reducing the risk of drawing conclusions from unrepresentative samples.

Potential artifacts due to the threshold methodology chosen discussed in previous point: It is not clear whether some of the patterns in the maps (e.g. the higher JJA values over Ravenna in Fig. 8 AvIn and Fig. 9 PkIn) are genuine results or artifacts of the methodology filtering out structures with higher thresholds (higher thresholds over Ravenna in Fig. 1). Line 257: Are the "hotspots" results of higher local thresholds? Or an actual signal? Same for "isolated spots" line 278.

Thank you for raising this question. Some of the patterns you highlight, such as the JJA intensity signal near Ravenna, are not artifacts of the thresholding methodology but features of the reanalysis, as also documented in previous validation studies (Viterbo et al., 2024; Cavalleri et al., 2024b; Giordani et al., 2025). In these areas, summertime convection tends to be excessively triggerer, leading to large rainfall amounts. This mechanism, already discussed in the revised Discussion section of our manuscript, explains the presence of such localized "hotspots". The fact that this is not a result of varying threshold is also supported from the fact that an analysis analogous to Figure 8, but based on a fixed 1 mm threshold, produced very similar spatial patterns (figure S4 in Supplementary Material):

[Figure]

We will clarify this point in the manuscript to make explicit that these signals originate from the reanalysis itself rather than from the structure identification procedure.

Cavalleri, F., C. Lussana, F. Viterbo, M. Brunetti, R. Bonanno, V. Manara, M. Lacavalla, S. Sperati, and M. Raffa (2024). Multi-scale assessment of high-resolution reanalysis precipitation fields over Italy. *Atmospheric Research*, 312, 107734. https://doi.org/10.1016/j.atmosres.2024.107734

Viterbo, F., S. Sperati, B. Vitali, F. D'Amico, F. Cavalleri, R. Bonanno, and M. Lacavalla (2024). MERIDA HRES: A New High-Resolution Reanalysis Dataset for Italy. *Meteorological Applications*, 31(6), e70011. https://doi.org/10.1002/met.70011

Giordani, A., P. Ruggieri, and S. Di Sabatino (2025). Added value of a multi-model ensemble of convection-permitting rainfall reanalyses over Italy. *Atmospheric Research*, 328, 108402. https://doi.org/10.1016/j.atmosres.2025.108402

The role of the "minimum enclosing ellipse" is unclear. Its mention in the title of Fig. 2 and in the text (line 169) is confusing: are the variables in Table 1 computed on the ellipse itself or on the rain object it encloses? Is the ellipse only used for visualization of selected structures, or does it play a role in extracting the properties of the rain objects? This part should be clarified for better understanding.

Thank you for pointing out this source of confusion. The minimum enclosing ellipse is used to compute the maximum spatial extent of each rainfall structure: specifically, the major axis of the ellipse provides a linear measure of event size. This approach is motivated by both operational and research applications (e.g., Wernli et al. (2008); SAL method), where representing an event as an ellipse allows extraction of its key spatial characteristics in a consistent way. The ellipse is therefore not only a visualization tool but also plays a role in defining one of the descriptors listed in Table 1 (axis_maj). In the following comment on spatial scale definition, we further justify why we chose maximum event extent—rather than area—as the most informative metric for our purposes.

Line 169: for readers unfamiliar with Wernli et al. (2008), it would be helpful if the purpose of using this methodology were explained.

In the revised manuscript, we will add a short explanation of the methodology by Wernli et al. (2008) to guide readers who may not be familiar with it, clarifying its purpose and relevance for our analysis.

Following the previous points, it is unclear whether axis_maj refers to the major axis of (i) the rain structure itself or (ii) the enclosing ellipse. In both cases, the phrasing in line 264 ("summer events are generally smaller") is problematic, as are later formulations in the SpS analysis regarding size. If (i), and the aim is to characterize the size of the structures, the "area" variable would be more appropriate. For example, a thin elongated structure may be characterized as large with SpS even though its area of coverage is small, while a rounder

structure with the same SpS value could cover a much larger area, yet still be classified as the same "size" when using SpS. If (ii), the analysis would instead describe the size of influence of the structure rather than its actual size. It should therefore be clarified what SpS is intended to capture, and statements about "size" should be made with caution. Replacing the SpS analysis with an analysis of the area variable would make the interpretation more straightforward.

We agree with the reviewer that the definition of event "size" was not sufficiently clear in the original manuscript. The *axis_maj* variable refers to the major axis of the minimum enclosing ellipse (ii), which we use as a consistent measure of the maximum extension of each event. In the revised manuscript, we will therefore replace the term *size* with *maximum extension of the event*. As you suggest, what we aim to capture is the *size of influence* of the structure rather than its area. We chose this linear metric over area because atmospheric phenomena are commonly characterized in terms of linear spatial scales (e.g., meso-beta, meso-gamma) expressed in kilometers, rather than in square kilometers (Thunis and Bornstein, 1996). Formulations such as line 264 will be revised to read, for example: "summer events generally have a smaller maximum spatial extension."

Thunis, P., and Bornstein, R. (1996). Hierarchy of Mesoscale Flow Assumptions and Equations. *Journal of Atmospheric Sciences*, 53, 380–397. https://doi.org/10.1175/1520-0469(1996)053<0380:HOMFAA>2.0.CO;2

**Technical comments**

- Line 10: "the most extreme component." The word component is unclear here. I suggest rephrasing to: "The most extreme rain events (EPEs)…"

We agree that the wording was unclear. What we intended to convey was the most extreme subset of the identified rainfall structures. In the revised manuscript, we will rephrase this expression for clarity, adopting a formulation such as "the most extreme rain events (EPEs)."

- Line 111: "event-based approach." As discussed earlier, this would be better described as an approach using hourly rain spatial structures. I also suggest adapting the phrase to be less generic and more explicit about how clustering helps address the limitations: "In light of these limitations, an object-based approach using a clustering technique was adopted to capture coherent hourly precipitation structures and reduce sensitivity to small-scale discrepancies between simulations and observations."

In the revised manuscript, we will describe our methodology as an "approach based on hourly precipitation spatial structures", rather than using the generic term "event-based approach." We will also be more explicit about the clustering technique employed, while directing the reader to the Methods section for the detailed explanation.

- Line 146: "The event-detection … precipitation events…" The vocabulary of event throughout the text before line 164 is misleading, since line 164 specifies: "Hereafter, the term 'event' denotes the precipitation structures identified using this method." Event terminology should not appear earlier without this clarification.

As clarified in our previous responses, in the revised manuscript we will consistently refer to the identified clusters as "Hourly Rain Spatial Structures", rather than "events".

- Line 296: "is conducted on a subset of EPEs." A clearer phrasing would be: "a subset of the HOPE-X dataset." (But HOPE-X should be renamed)

We agree that this phrasing is clearer. In the revised manuscript, the dataset will be renamed.

- Lines 321–322: The phrase "seasonal redistribution is likely driven by the persistence of summer-like convective activity into early autumn" is confusing, since the focus is on differences between summer and autumn. This cannot be justified by describing autumn activity as "summer-like." A phenomenon-based explanation could instead highlight that the persistence of warm sea surface conditions beneath a cooler atmosphere creates instability favorable to convection. Autumnal convection is not "summer-like". This is also reflected in Fig. 12, which shows differences in the "size" characteristics of hourly rain structures between summer and autumn.

We agree with your phenomenon-based interpretation, which will be incorporated in the revised manuscript. Confusing expressions such as "summer-like" will be removed, and we will highlight that the persistence of warm sea surface conditions beneath a cooler atmosphere creates instability favorable to convection. We will also use information on the maximum extension of events (Fig. 12) to emphasize both similarities and differences between summer and autumn convective activity. We appreciate this useful suggestion.

- Line 354: Instead of "alternating pattern," I suggest: "spatial heterogeneity in the sign of the signal" or "patterns of alternating signs."

We agree with this suggestion and will revise the manuscript to replace "alternating pattern" with "spatial heterogeneity in the sign of the signal" for clarity.

---

## Author Response (AR1)

**Answers to the reviews of manuscript egusphere-2025-3455**
**submitted to Natural Hazards and Earth System Sciences (NHESS)**

**Anonymous Referee #1, 28 Aug 2025**

[answers in blue]

The manuscript presents reanalyses model simulation results on convective-permitting scales over Italy with hourly resolution. The study is important since understanding precipitation and provides a method of studying events at hourly scales. Extreme precipitation trends on an hourly basis are important for assessing flood risk, especially as these trends are expected to increase with a warming climate, even in drying areas. The paper is generally well written, and the figures are well presented, however there are a few major concerns, along with some minor comments, that I hope the authors will address in order for the article to be suitable for publication.

**General comments**

A major concern is that the title and abstract of the paper describe extremes, while a lot of the results and figures describes the full dataset compared to the extreme analysis. The Result section should be altered to highlight the extreme analyses better and maybe reduce the description and figures around the full dataset to better suit the journal. Another major concern is that with the weight given on extremes in titles and abstract, there is only one threshold for extremes used which also is set very high, especially when considering hourly data. Above the mean of 37 datapoints which give only 18-19 events per grid cell if the events are evenly distributed around the mean (a quick calculation sets this threshold around the 9.995 percentile). The threshold applied to define extreme precipitation is exceptionally high, and with so few events included, the resulting trend estimates are highly uncertain. The robustness of the study would be improved if additional analyses were carried out using several lower thresholds, allowing for a more comprehensive assessment of trends.

We would like to sincerely thank the reviewer for the careful and thorough reading of our manuscript and for the constructive comments, which will greatly help us improve the quality of the paper.

**Response to Major Concern 1:** We acknowledge that the original *Results* section devoted substantial space to the analysis of the full dataset. Our intention was to demonstrate that the identified precipitation events are consistent with the climatological characteristics of hourly precipitation over Italy, thereby providing a basis for the subsequent analysis of extremes. However, we agreed that the section needed better alignment with the paper's focus on extremes. In the revised manuscript, we removed Figure 9 and shortened Section 3.1 to give greater emphasis to the analyses of extreme events.

**Response to Major Concern 2:** It is true that the applied threshold corresponds to approximately 18–19 events per grid cell if events were selected considering single grid points independently. However, events were not selected point by point: we used a

*structure-based* approach, where a cluster of grid points is classified as a precipitation structure belonging to the class of extremes if at least one grid point within the cluster exceeds the threshold. This approach results in a less restrictive selection than it may appear. For instance, about 11% of summer events exceeded the average RX1hour threshold (corresponding to about 4000 extremes each summer inside the full domain, see Fig. 11 for their spatial distribution), consistent with the seasonal intensity variability (see revised Section 3.1).

We recognize that lowering the threshold would increase the number of extremes, but would also reduce interpretability by mixing extremes (in the meaning of Extreme Value Theory, Coles, 2001) with high-quantile events. Our use of a fixed threshold based on the mean of annual maxima (RX1hour) follows the rationale adopted by Lavers et al. (2025), who recommended fixed thresholds (RX1day) for clarity in communicating extremes. A clearer explanation of these methodological choices has been added in Section 2.4 (lines 214–220) of the revised manuscript.

We also addressed potential robustness concerns by applying spatial aggregation (0.5° × 0.5°) and trend detection methods including Sen's slope, Mann–Kendall testing, and False Discovery Rate control (Wilks, 2006), ensuring reliability despite the limited sample size (see lines 238-244)

Coles, S. (2001), An Introduction to Statistical Modeling of Extreme Values, Springer-Verlag, London, https://link.springer.com/book/10.1007/978-1-4471-3675-0

Wilks, D. S., 2006: On "Field Significance" and the False Discovery Rate. J. Appl. Meteor. Climatol., 45, 1181–1189, https://doi.org/10.1175/JAM2404.1

**Specific Comments**

Page 2. L27-29: Even drying areas experience more extreme precipitation events.

We thank the reviewer for this useful remark. In the revised manuscript (lines 27-28), we clarified this point by explicitly acknowledging that even regions undergoing drying trends may experience an intensification of extreme precipitation events, as noted by Donat et al. (2016).

Donat, M., Lowry, A., Alexander, L., O'Gorman, P., & Maher, N. (2016). More extreme precipitation in the world's dry and wet regions. *Nature Climate Change*, 6, 508-513. https://doi.org/10.1038/NCLIMATE2941

Page 5. Section 2.2  Median is easier than the 50th percentile, and used earlier in Introduction.

We appreciate the reviewer's suggestion. We agreed that the term *median* is clearer and more consistent with its earlier use in the Introduction. In the revised manuscript, we replaced "50th percentile" with *median* at line 161 and throughout the text, retaining "(i.e., 50th percentile)" only at its first occurrence to clarify the statistical meaning.

The Method section needs to be improved to better understand the results presented. Section 2.2 should be rewritten to increase readability. The thresholds and smoothing are first presented, and then described again in the later paragraph, maybe rewrite for better readability.

We fully acknowledge the reviewer's comment regarding the need to improve the readability of Section 2.2. In the revised manuscript, we restructured this section (lines 159–191) to present the thresholds and their smoothing together in a single, coherent sequence, thereby eliminating repetition and improving clarity. The section was also revised in accordance with comments from other reviewers, unifying all considerations related to threshold definition and smoothing choices.

Line 162 – 165 More specifically, … this sentence in especially hard to follow. More than half in which instances?

We agree with the reviewer that the original formulation was difficult to follow. In the revised manuscript, we rephrased the sentence to make the definition clearer and more precise (lines 179–180).

I'm also curious how time is handled as an event usually lasts more than one hour. Is there any clustering in time?

We thank the reviewer for raising this important point. In our analysis, the general approach was to study events at the hourly scale, regardless of what happens before or after. However, the aspect of event duration is explicitly addressed in the final part of Section 3.3, where we analyze the average persistence of extreme events (Figure 14). This analysis provides an estimate of the mean duration of the extremes as defined in our study and shows that such events typically last only slightly longer than one hour on average. This finding clarifies the temporal scale of our definition of extremes and supports the robustness of the calculated trends by minimizing the risk of double counting. In the revised manuscript, we stated this choice more clearerly (lines 111–113 and 230).

Section 2.4 Is this Gaussian filter the same as used in section 2.2? If so, does the justification of this radius apply to the earlier smoothing, and move this part in section 2.2?

We thank the reviewer for this observation. In the revised manuscript, the explanation was unified in Section 2.2 (lines 171-174).

Table 2, The table could be improved if the names were included in addition to the short names.

We thank the reviewer for this helpful suggestion. In the revised manuscript, we added a column with the full names alongside the short names to improve readability (see revised Table 2).

Page 9. L196 ERA5 is already introduced as the driver of the dataset.

We thank the reviewer for this remark. In the revised manuscript, we removed the phrase "which is a global reanalysis product with a 0.25° grid spacing" to avoid repeating information already introduced when describing the ERA5 dataset.

Page 10, l207-208 Explain better here: How many events would this give over a typical grid cell on the coast or in the mountains?

We appreciate the reviewer's request for clarification. As discussed in response to the second major concern, extremes are not selected point by point. Instead, we use an event-based approach, where an event is defined as a cluster of grid points and classified as an Extreme Precipitation Event (EPE) if at least its peak exceeds the threshold. Precipitation distributions are typically right-skewed with heavy tails and vary seasonally and regionally, leading to more extremes than the 17–18 events expected from point-by-point selection and substantial variability across regions (e.g., coastal vs. mountainous areas), as shown in Figure 11. The number of occurrences is computed within a 0.5° moving window (≈156 grid points) rather than at single grid cells to reduce misplacement errors. In the revised manuscript, we clarified this at lines 196–198.

Page 20. L 358-359: This analysis would also benefit from a lower threshold, as two executive hours above RX1hour is extremely rare.

We agree with the reviewer that two consecutive hours above the average RX1hour are quite rare. However, this rarity is consistent with our methodological choice: extremes are defined from the mean of hourly annual maxima (RX1hour) to identify only the most exceptional events, in line with Extreme Value Theory. This makes consecutive exceedances unlikely but confirms that the analysis truly focuses on extremes and supports treating extremes of different hours as approximately independent. As noted in our response to the second major concern, lowering the threshold would increase the number of events but blur the distinction between genuine extremes and high-quantile precipitation, and would require multi-hour event tracking, which is beyond the scope of this work. In the revised manuscript, we clarified this point at lines 215-220.

Page 22. L 423-424, This sentence could be misinterpreted, there could be trends here that were not found because of issues with the reanalysis. Could there be a false positive trend, or a false insignificant trend?

We thank the reviewer for this important observation. We agree that limitations of the reanalysis data may have masked existing trends, potentially leading to false insignificant trends—i.e., trends that are present but not robustly detectable due to dataset constraints. In principle, such biases could have masked decreasing trends in those areas; however, the overall spatial pattern suggests that this scenario is highly unlikely. What we intended to convey is that the significant trends we do identify remain robust despite these limitations. In the revised manuscript, we clarify this point at lines 413–414.

**Anonymous Referee #2, 1 Sep 2025**

**General comments:** This manuscript presents an analysis of precipitation event properties (frequency, intensity, duration, and size) using a convection-permitting reanalysis dataset for Italy. The paper is well written, scientifically rigorous, and of clear interest to the NHESS community. That said, the manuscript would benefit from more cautious phrasing regarding the ability of the reanalysis to represent meso-beta scale processes, a more concise and focused introduction, and more consistent use of acronyms. These revisions would improve readability and balance some of the claims. I therefore recommend major revisions before the paper can be considered for publication.

**Specific comment:** I find the manuscript highly relevant, as it combines two approaches that are still not very common in the community: the high-resolution reanalysis from ERA5 and WRF, and the event-based approach. This framework could be extended to other regions, which increases the value of the study. But the paper would benefit from addressing the following points:

- Introduction: The introduction is overly long. Please consider streamlining it to improve readability and focus on the main motivation and novelty of the study.

We thank the reviewer for this constructive suggestion. In the revised manuscript, we streamlined the Introduction to improve readability and better highlight the main motivation and novelty of the study. In particular, we shortened the review of previous works and moved the description of MERIDA HRES biases to Section 2.1. The revised Introduction now accounts for 95 lines out of 460 (≈20% of the paper, excluding references), which we believe represents a good compromise between completeness and conciseness.

- Section 2.1: Here you present the MERIDA HRES reanalysis, but you do not describe the quality or limitations of the dataset. While some of this is discussed later, it would be more appropriate to include a clear description of dataset strengths and limitations already in this section.

We agree that Section 2.1 would benefit from a clearer description of the strengths and limitations of the MERIDA HRES reanalysis. While these aspects have been discussed in our previous works (Cavalleri et al., 2024; Viterbo et al., 2024), in the revised manuscript we expanded Section 2.1 (lines 136-157) to provide a concise overview of the dataset's quality, highlighting both its ability to capture precipitation processes at convection-permitting scales and its known limitations, by moving some sentences from the Introduction and the Discussion.

Cavalleri, F., C. Lussana, F. Viterbo, M. Brunetti, R. Bonanno, V. Manara, M. Lacavalla, S. Sperati, and M. Raffa (2024). *Multi-scale assessment of high-resolution reanalysis precipitation fields over Italy*. Atmospheric Research, 312, 107734. https://doi.org/10.1016/j.atmosres.2024.107734

Viterbo, F., S. Sperati, B. Vitali, F. D'Amico, F. Cavalleri, R. Bonanno, and M. Lacavalla (2024). *MERIDA HRES: A New High-Resolution Reanalysis Dataset for Italy*. Meteorological Applications, 31(6), e70011. https://doi.org/10.1002/met.70011

- Resolution: The reanalysis resolution is reported as 4 km, presumably referring to the horizontal grid spacing. What about the vertical resolution? A more careful discussion of its role would be valuable. In addition, in the results you claim that the model can resolve meso-beta processes (2–20 km). In my view, a 4 km horizontal resolution may not be sufficient to adequately resolve the full meso-beta range, perhaps only processes above ~10 km. For example, at 4 km resolution, a 10 km feature would be represented by only a handful of grid points. I recommend revising this claim to reflect these limitations.

We thank the reviewer for this important observation. We agree that additional detail on the resolution of MERIDA HRES is needed. In the revised manuscript (lines 127–128), we explicitly state that MERIDA HRES has 56 vertical levels, with increased vertical resolution in the lower atmosphere (e.g., levels at 10, 35, 70, 100, 130, 180, 250, 325, 415, and 500 m; Viterbo et al., 2024). The effective horizontal resolution has been evaluated in previous work using a wavelet spectral decomposition approach (Cavalleri et al., 2024b), which demonstrated the dataset's ability to represent convective precipitation events. At the same time, we revised our discussion of meso-beta processes to provide a more balanced assessment at lines 139-141 and 268–271, and expanded this discussion at lines 397–401.

Viterbo, F., S. Sperati, B. Vitali, F. D'Amico, F. Cavalleri, R. Bonanno, and M. Lacavalla (2024). *MERIDA HRES: A New High-Resolution Reanalysis Dataset for Italy*. Meteorological Applications, 31(6), e70011. https://doi.org/10.1002/met.70011

- Acronyms: The manuscript contains many acronyms, which makes it easy to lose track. I recommend reducing their use where possible and writing terms explicitly. I will provide examples in the technical corrections.

We thank the reviewer for this helpful suggestion. In the revised manuscript, we carefully reviewed the text to reduce the number of acronyms and wrote terms explicitly where possible to improve readability. In particular, we replaced *AvIn*, *PkIn*, and *SpS* with the more intuitive *MeanInt*, *PeakInt*, and *SpatExtent*, and generally limited acronym use while keeping key terms clear and consistent throughout the paper.

**Technical corrections:**

-Line 4: Please specify explicitly how many years of data are available.

Thank you for pointing this out. In the revised manuscript, we explicitly stated that the dataset spans 37 years, from 1986 to 2022 (line 3).

-Line 25: Here you state that extreme precipitation changes are due to thermodynamics (CC scaling). What about the role of dynamics? For instance, see Pfahl et al. (2017):

Pfahl, S., O'Gorman, P. A., & Fischer, E. M. (2017). Understanding the regional pattern of projected future changes in extreme precipitation. Nature Climate Change, 7(6), 423-427.

We thank the reviewer for this important comment and for suggesting a valuable reference. We agree that changes in extreme precipitation are influenced not only by thermodynamic effects, such as Clausius–Clapeyron scaling, but also by dynamic processes. While thermodynamic effects generally drive a uniform fractional increase in extremes, dynamic factors can modulate these changes regionally, affecting both intensity and frequency (Pfahl et al., 2017). In the revised manuscript, we clarified this point and discussed both thermodynamic and dynamic influences on extreme precipitation (lines 30-32).

- Line 51: Please, define ARCIS.

Thank you for pointing this out. In the revised manuscript, we explicitly defined ARCIS at line 54 as the *"Archivio Climatologico per l'Italia Centro Settentrionale"*, a high-resolution climate precipitation dataset for north-central Italy (Pavan et al., 2019).

Pavan, V., Antolini, G., Barbiero, R., Berni, N., Brunier, F., C., A. Cagnati, O. Cazzuli, A. Cicogna, C. De Luigi, E. Di Carlo, M. Francioni, L. Maraldo, G. Marigo, S. Micheletti, L. Onorato, E. Panettieri, U. Pellegrini, R. Pelosini, D. Piccinini, S. Ratto, C. Ronchi, L. Rusca, S. Sofia, M. Stelluti, R. Tomozeiu, T. Torrigiani Malaspina, 2019: *High resolution climate precipitation analysis for north-central Italy, 1961–2015.* Clim. Dyn. 52, 3435–3453. https://doi.org/10.1007/s00382-018-4337-6

- Line 65: Please, define GRIPHO

In the revised manuscript, we explicitly defined GRIPHO at line 66 as *"GRidded Italian Precipitation Hourly Observations"*, a high-resolution precipitation dataset for Italy (Fantini, 2019).

Fantini, A. *Climate change impact on flood hazard over Italy*, PhD Thesis, (2019).

-Lines 77-79: Please add supporting references.

We thank the reviewer for this suggestion. In the revised manuscript, we added supporting references at lines 75–79, explaining that operative rain gauge networks tend to underestimate extreme convective precipitation over small areas, with point-scale extremes underestimated by about 20% (Schroeer et al., 2018), while radar- and satellite-based precipitation estimates may show large positive or negative biases (Wang et al., 2021).

Schroeer, K., Kirchengast, G., & O, S. (2018). *Strong Dependence of Extreme Convective Precipitation Intensities on Gauge Network Density.* Geophysical Research Letters, 45, 8253–8263. https://doi.org/10.1029/2018GL077994

Wang, S., Li, C., Li, D., Tian, X., Bao, H., Chen, G., & Xia, Y. (2021). *Exploring the utility of radar and satellite-sensed precipitation and their dynamic bias correction for integrated prediction of flood and landslide hazards.* Journal of Hydrology. https://doi.org/10.1016/j.jhydrol.2021.126964

-Line 147: How sensitive are your results to the choice of the 50th percentile threshold?

We thank the reviewer for this comment. We tested the sensitivity of our results by constructing alternative datasets using different thresholds, both percentile-based and fixed. In particular, using a fixed 1 mm threshold (see Supplementary Material), our analyses show

only minor variations, indicating that the observed seasonal differences mainly reflect genuine variability rather than artifacts from the clustering method. This also holds for extreme precipitation trends, confirming that our results are robust to the choice of the 50th percentile threshold. We commented on this explicitly in the revised manuscript at lines 315–319.

-Figure 2: Consider using a single colorbar and enlarging it for clarity.

We thank the reviewer for this helpful suggestion. In the revised manuscript, we adjusted Figure 2 to use a single, enlarged colorbar to improve clarity.

-Table 1: Consider alternative acronyms instead of axis_maj or lon_wavg, lat_wavg, which are not very intuitive.

We thank the reviewer for this helpful suggestion. In the revised manuscript, we updated the acronyms in Table 1, replacing axis_maj, lon_wavg, and lat_wavg with the more intuitive max_extent, cdm_lon, and cdm_lat, in line with other reviewers' observations on the actual meaning of the spatial extent of a structure.

-Line 178: Have you considered that grid cell size decreases with latitude? This could affect your area-based results.

Thank you for the comment. We have considered the variation of grid cell size with latitude. Across our study area (36°N–47°N), the relative difference in grid cell area due to meridian convergence is limited, with cells at higher latitudes being slightly smaller. To quantify this effect, we used the NOAA distance calculator (https://www.nhc.noaa.gov/gccalc.shtml) and verified that the distance represented by 0.1° in longitude decreases only modestly with latitude: for example, from approximately 9 km at 36°N to about 8 km at 47°N.

For many large-scale analyses, this difference (on the order of ~10–15%) is considered negligible, particularly when results are aggregated or averaged across the domain. Our objective is to provide a consistent view of the relationship between spatial scales and the characteristics of the identified events. In this context, the modest variations in grid cell shape and size do not significantly affect our conclusions.

 -Line 181-182: The acronyms here are difficult to follow, please consider spelling them out.

We thank the reviewer for this comment. In the revised manuscript, we followed this suggestion and avoided using acronyms there. More generally, we replaced several acronyms with clearer ones or spelled them out where appropriate to improve readability, as also recommended by Reviewer #1.

-Line 230: Please clarify whether this refers to a large quasi-stationary cyclone.

We thank the reviewer for this comment. You are correct that this refers to a large quasi-stationary cyclone. In the revised manuscript, we clarified this explicitly at line 257.

-Line 240: Be cautious with the claim that 4 km simulations can fully resolve these processes; please consider rephrasing.

We thank the reviewer for this comment. We agree that the original statement was too strong. In the revised manuscript, we rephrased it more cautiously to state that 4 km simulations can *explicitly represent* convection and related processes, but may not *fully resolve* them. This revised wording more accurately reflects the capabilities and limitations of convection-permitting models. The clarification was made at lines 139-141 and 268–271, and expanded this discussion at lines 397–401.

-Line 374-377: It would be useful to mention dataset biases earlier (e.g. in the data section), and to quantify them explicitly as percentages relative to observations.

We thank the reviewer for this comment. In the revised manuscript, we introduced the discussion of dataset biases earlier in Data Section 2.1 (lines 149–157) and provided a quantitative estimate of these biases as percentages relative to observations, also in line with Reviewer #1's suggestion.

-Line 378: Please elaborate on the role of skin temperature in WRF biases.

We thank the reviewer for this comment. In the revised manuscript, we elaborated on the role of skin temperature in WRF biases, explaining how deviations in surface temperature can propagate to near-surface variables and influence model performance. This clarification was added at lines 402–407.

**Anonymous Referee #3, 19 Sep 2025**

[answers in blue]

**General comments**

The authors make use of a high-resolution reanalysis dataset that represents a considerable effort in terms of length (37 years), spatial coverage (all of Italy), and resolution (4 km). This combination makes it a valuable resource for a national-scale study of changes in extreme precipitation. The availability of hourly rainfall data also makes the study relevant for analyzing the seasonality and spatial variability of hourly precipitation in Italy.

I consider this work useful for the scientific community because it brings together results that were previously limited to local or regional scales into a study covering the whole of Italy. This allows for a better understanding of the spatial variability of extreme precipitation event (EPE) characteristics and their changes, while showing consistency with earlier regional studies.

That said, I find that the Abstract and Introduction could be more explicit regarding the specific scientific contributions and aims of the work. The chosen methodology is interesting in that it introduces a structure-based perspective on hourly precipitation; however, some methodological aspects would benefit from clearer explanation or reconsideration.

Referring to hourly precipitation structures as "events" is, in my view, misleading. Since their temporal evolution (duration, displacement, deformation) is not considered, nor the total rainfall volume produced by multi-hour storm systems, the term "event" may create confusion and should be replaced with a more precise designation. Finally, the section addressing non-extreme precipitation should be better introduced, with its purpose and relevance clarified.

Thank you very much for your careful reading of our manuscript and for your constructive comments, which have been very helpful in improving the quality and clarity of our work.

We have revised the Abstract and Introduction to make the scientific contributions and aims of the study more explicit. The methodology section has also been improved for greater clarity. In particular, following your suggestion, we have replaced the term *"event"* with *"Hourly Precipitation Spatial Structure"* to avoid confusion and to better reflect the nature of the analyzed objects. Furthermore, the section discussing non-extreme precipitation has been shortened and its purpose clarified, emphasizing its role in contextualizing the subsequent analysis of hourly precipitation extremes.

Detailed references to the specific changes made in response to each of your comments are provided in the following point-by-point responses as well as in the tracked-changes manuscript. We sincerely thank you again for your valuable review and helpful feedback.

Overall, in my opinion, the manuscript is suitable for publication after minor revisions. These include adjustments in specific vocabulary, clearer argumentation, additional explanations to help reader understanding, and refinements in the methodology.

**Specific comments**

The 4 km resolution of the reanalysis is at the limit of what is usually called convection-permitting. This should be made explicit in the text, especially since the dataset paper uses "high-resolution" instead. I would keep the term "convection-permitting" for clarity but suggest adding a note that it is at the edge of the definition.

We thank the reviewer for this valuable suggestion. In the revised manuscript, we consistently refer to MERIDA HRES as a *convection-permitting* reanalysis rather than *high-resolution*. As noted, its 4 km horizontal spacing lies at the boundary of the convection-permitting definition, and we have made this explicit at lines 139-141 and 268–271, and further discussed at lines 397–401. This clarification improves transparency regarding the dataset's capabilities and limitations (Viterbo et al., 2024; Cavalleri et al., 2024b).

Cavalleri, F., C. Lussana, F. Viterbo, M. Brunetti, R. Bonanno, V. Manara, M. Lacavalla, S. Sperati, and M. Raffa (2024). *Multi-scale assessment of high-resolution reanalysis precipitation fields over Italy*. Atmospheric Research, 312, 107734. https://doi.org/10.1016/j.atmosres.2024.107734

Viterbo, F., S. Sperati, B. Vitali, F. D'Amico, F. Cavalleri, R. Bonanno, and M. Lacavalla (2024). *MERIDA HRES: A New High-Resolution Reanalysis Dataset for Italy*. Meteorological Applications, 31(6), e70011. https://doi.org/10.1002/met.70011

Since about half of the figures in the Results section concern hourly rain structures and not EPEs, it is important to either (i) include non-extreme hourly rain discussion in the abstract as well as adapt the title accordingly, or (ii) introduce non-extreme hourly rain results as a necessary step before moving to extremes. Otherwise, the part on extremes, which is announced to the reader, takes too long to arrive.

We thank the reviewer for this insightful comment. In the revised manuscript, we address this in two ways: (i) we now mention in the Abstract (lines 7–8) the findings related to non-extreme hourly precipitation structures, and (ii) we clarify at the beginning of Section 3.1 (lines 248–249) the motivation for including the analysis of non-extreme rainfall as a necessary step toward understanding extremes. Following this suggestion, and in line with Reviewer 1's comments, we have also shortened the discussion of non-extreme precipitation so that the focus on extremes—highlighted in the title and Abstract—emerges earlier in the Results section.

The Introduction and Abstract should state more clearly the purpose of the work and its scientific contribution.

We thank the reviewer for this valuable suggestion. In the revised manuscript, we have clarified the purpose and scientific contribution of the study more explicitly. Specifically, we

adapted a sentence in the Abstract (lines 18–20) highlighting that the goal of the work is to advance understanding of extreme precipitation trends in Italy through an innovative approach based on hourly fields from a convection-permitting reanalysis. In addition, we expanded the corresponding paragraph in the Introduction (lines 82–84) to clearly state the study's objectives and its contribution to the broader discussion on the characterization of hourly precipitation extremes.

The HOPE-X dataset is a collection of hourly rain structures, not "events" in the usual sense of storm systems or local rain events. For instance, a precipitation system lasting two hours can be counted as two "events," even though it is the same system. Similarly, a moving storm can be counted as several "events" in different areas as it displaces. For this reason, the use of the term "events" is inappropriate. The title is also misleading, since the study is not, in my opinion, event-based. The same applies to the dataset name HOPE-X. This does not invalidate the study, but the terminology issue is central and should be revised. I strongly recommend using the term "hourly rain structures." If the term "event" is retained, it should not appear in the title, abstract, or in any part of the text where the definition is not clearly introduced yet (line 164: "Hereafter, the term 'event' denotes the precipitation structures identified using this method"). In that case, it should also be stated that the term "event" is used for readability purposes only.

We thank the reviewer for this comment. We fully agree that the term *"event"* may be misleading in the context of our analysis, since the HOPE-X dataset is composed of hourly precipitation spatial fields rather than temporally continuous storm systems. In the revised manuscript, we have therefore adopted more precise terminology throughout. Specifically, we now refer to *Hourly Precipitation Spatial Structures (HPSSs)* instead of *"events"*, and to *Hourly Precipitation Extremes (HPEs)* instead of *"Extreme Precipitation Events (EPEs)"*. The dataset has accordingly been renamed HOPSS-X (*HOurly Precipitation Spatial Structures and eXtremes*). We also modified the title accordingly.

At the same time, we retain limited references to *"extreme events"* in specific contexts where this term is already well established in the literature, clarifying that in an Eulerian framework (e.g., Ignaccolo & De Michele, 2010), "event" denotes the occurrence of an HPSS within a fixed spatial reference rather than a tracked, temporally evolving storm. This distinction and the rationale for our terminology are now clearly stated in the revised manuscript (lines 229–234).

Moreover, we expanded the discussion to emphasize that HPEs are typically short-lived—usually confined to a single hour—thus maintaining consistency with the concept of an "extreme event" in an Eulerian sense. This is supported by the analysis of HPE persistences (see Fig. 14 and discussion). To further support this point, we also provide in this response the distribution of HPE durations within 0.1° windows for JJA and SON:

[Figure]

[Figure]

This underscores that HPEs almost always last only one hour and can therefore still be meaningfully associated with the concept of an "extreme event" in an Eulerian framework.

Ignaccolo, M. & De Michele, C., 2010. A point based Eulerian definition of rain event based on statistical properties of inter drop time intervals: An application to Chilbolton data. *Advances in Water Resources*, 33(8), pp.933–941. doi:10.1016/j.advwatres.2010.04.002

Following the previous comment, in maps such as Figure 6 the number of "events" (N) can reach ~300 per season in a 0.5×0.5 window. This N reflects a mix of the number of hours with a rain structure and the number of surrounding structures at each hour, rather than distinct events. For example, an hour with many small rain structures and an hour with one large rain structure could produce the same hourly rainfall total in that 0.5×0.5 area, yet N

would differ greatly. It should therefore either (i) be clarified in the text what the analysis of N is intended to represent if it is something other than the amount of rain, (ii) reconsider the counting methodology, or (iii) include additional analysis showing that N reflects the actual volume of hourly rain.

We thank the reviewer for this thoughtful comment and fully agree that clarification is needed regarding what the metric *N* represents. The purpose of this analysis is to identify areas where precipitation structures occur most frequently, quantified as the number of structure centers of mass within each 0.5° × 0.5° window. This metric reflects the frequency of occurrence of rain structures rather than the rainfall amount, which is instead captured by the mean intensity analysis.

To address the reviewer's concern, we also tested an alternative metric — the frequency of wet hours (i.e., the fraction of hours with at least one detected structure in the window). The following figure is also added in the supplementary material:

[Figure]

The resulting spatial distribution closely mirrors that of *N*, confirming that both measures convey similar information. In the revised manuscript, we clarify this aspect at lines 204–206.

The use of seasonal thresholds to adapt to the intensity scales of each season is well justified and explored. However, the choice to let the seasonal threshold vary spatially with the local 50th percentile of 1 mm+ hourly rains requires stronger justification. As a result of threshold methodology, identical rain structures may be included in HOPE-X in one region

but not in another. While such area-relative thresholds are understandable when defining extremes (e.g. EPEs) due to the definition of rarity being potentially region-based, they are harder to justify for non-extreme rain structures. Choosing fixed threshold across the domain for each season would be more correct. If the authors choose to go with a justification of the current methodological choice of non-homogeneous threshold, consider also that in areas with very little rainfall, the threshold based on wet hours 1mm+ may also be computed on a very small and potentially unrepresentative sample, so this would need to be tested.

We thank the reviewer for this detailed and insightful comment. The choice of a spatially varying, seasonally dependent threshold was carefully evaluated through extensive sensitivity tests (included in the Supplementary Material), comparing both fixed (0.5, 1, 2 mm) and percentile-based (50th, 75th, 90th, 99th) approaches.

We ultimately adopted the spatially varying 50th-percentile threshold (computed over wet hours ≥ 1 mm) because precipitation regimes in Italy differ markedly by both season and region due to strong geographic contrasts (coastal vs. inland, mountain vs. plain, north vs. south). This adaptive threshold ensures that the identified precipitation structures remain meteorologically meaningful in all contexts—capturing comparable intensity scales across diverse climates—while avoiding overmerging in wetter areas and the detection of noise in drier ones. In other words, it provides a locally relevant and dynamically consistent definition of "significant precipitation."

We acknowledge, however, that in regions with scarce rainfall (e.g., over the sea in summer), the sample size for computing percentiles may be limited. This issue is mitigated because such regions exhibit very few structures overall and are excluded from the trend analysis (grey areas in Fig. 12), preventing the use of unrepresentative data.

In the revised manuscript, we clarify and justify this methodological choice in lines 164–171, explaining its advantages, limitations, and the tests supporting its robustness.

Potential artifacts due to the threshold methodology chosen discussed in previous point: It is not clear whether some of the patterns in the maps (e.g. the higher JJA values over Ravenna in Fig. 8 AvIn and Fig. 9 PkIn) are genuine results or artifacts of the methodology filtering out structures with higher thresholds (higher thresholds over Ravenna in Fig. 1). Line 257: Are the "hotspots" results of higher local thresholds? Or an actual signal? Same for "isolated spots" line 278.

We thank the reviewer for this important observation. The localized patterns highlighted—such as the enhanced JJA intensity near Ravenna—are not artifacts of the spatially varying threshold methodology but reflect features of the reanalysis, consistent with known model behavior. Previous evaluations of MERIDA HRES (Viterbo et al., 2024; Cavalleri et al., 2024b; Giordani et al., 2025) have documented a tendency for excessive convective triggering in this region during summer, resulting in locally amplified rainfall intensities. This interpretation is supported by some sensitivity tests: when repeating the analysis using a fixed 1 mm threshold, the resulting spatial patterns closely matched those in Figure 8 (see Supplementary Fig. S6, also reported in the following).

[Figure]

This confirms that the "hotspots" and isolated features observed (e.g., near Ravenna or in scattered coastal zones) originate from physical biases in the reanalysis, not from threshold-related artifacts.

We have clarified this point explicitly in the revised manuscript (lines 315–324), noting that these localized features correspond to known model tendencies rather than methodological effects.

Cavalleri, F., C. Lussana, F. Viterbo, M. Brunetti, R. Bonanno, V. Manara, M. Lacavalla, S. Sperati, and M. Raffa (2024). Multi-scale assessment of high-resolution reanalysis precipitation fields over Italy. *Atmospheric Research*, 312, 107734. https://doi.org/10.1016/j.atmosres.2024.107734

Viterbo, F., S. Sperati, B. Vitali, F. D'Amico, F. Cavalleri, R. Bonanno, and M. Lacavalla (2024). MERIDA HRES: A New High-Resolution Reanalysis Dataset for Italy. *Meteorological Applications*, 31(6), e70011. https://doi.org/10.1002/met.70011

Giordani, A., P. Ruggieri, and S. Di Sabatino (2025). Added value of a multi-model ensemble of convection-permitting rainfall reanalyses over Italy. *Atmospheric Research*, 328, 108402. https://doi.org/10.1016/j.atmosres.2025.108402

The role of the "minimum enclosing ellipse" is unclear. Its mention in the title of Fig. 2 and in the text (line 169) is confusing: are the variables in Table 1 computed on the ellipse itself or on the rain object it encloses? Is the ellipse only used for visualization of selected structures, or does it play a role in extracting the properties of the rain objects? This part should be clarified for better understanding.

We thank the reviewer for this helpful comment. We agree that the role of the *minimum enclosing ellipse* required clearer explanation. The ellipse is used to quantify the spatial

extent of each precipitation structure: specifically, its major axis defines the *maximum spatial extent* (previously *axis_maj*, now renamed *max_extent* in Table 1). This provides a consistent, orientation-independent linear descriptor of structure size. In the following comment on spatial scale definition, we further justify why we chose maximum event extent—rather than area—as the most informative metric for our purposes.

Line 169: for readers unfamiliar with Wernli et al. (2008), it would be helpful if the purpose of using this methodology were explained.

We thank the reviewer for this comment. In the revised manuscript, we kept the reference to *Wernli et al. (2008)* as an example of studies employing the minimum enclosing ellipse. We did not describe the full method, as it is not directly relevant here, but we briefly clarified at lines 186–187 the definition of minimum enclosing ellipse.

Following the previous points, it is unclear whether axis_maj refers to the major axis of (i) the rain structure itself or (ii) the enclosing ellipse. In both cases, the phrasing in line 264 ("summer events are generally smaller") is problematic, as are later formulations in the SpS analysis regarding size. If (i), and the aim is to characterize the size of the structures, the "area" variable would be more appropriate. For example, a thin elongated structure may be characterized as large with SpS even though its area of coverage is small, while a rounder structure with the same SpS value could cover a much larger area, yet still be classified as the same "size" when using SpS. If (ii), the analysis would instead describe the size of influence of the structure rather than its actual size. It should therefore be clarified what SpS is intended to capture, and statements about "size" should be made with caution. Replacing the SpS analysis with an analysis of the area variable would make the interpretation more straightforward.

We thank the reviewer for this insightful comment. We agree that the definition of event "size" required clarification. As correctly noted, this metric captures the spatial influence of the structure rather than its actual area. We chose this linear measure instead of area because atmospheric features are commonly described in terms of linear spatial scales (e.g., meso-beta, meso-gamma ranges; *Thunis and Bornstein, 1996*), rather than in square kilometers, as clarified in revised manuscript at lines 185–190. The old variable *axis_maj* refers to the major axis of the minimum enclosing ellipse, thus representing the maximum spatial extension of each structure. In the revised manuscript, we replaced it with the term *maximum spatial extension (max_extent)*, as defined at line 185 and used consistently throughout the text. Correspondingly, all formulations were revised for consistency — for example, at line 293 we now write: *"during summer, HPSSs have generally smaller extents."*

Thunis, P., and Bornstein, R. (1996). Hierarchy of Mesoscale Flow Assumptions and Equations. *Journal of Atmospheric Sciences*, 53, 380–397. https://doi.org/10.1175/1520-0469(1996)053<0380:HOMFAA>2.0.CO;2

**Technical comments**

- Line 10: "the most extreme component." The word component is unclear here. I suggest rephrasing to: "The most extreme rain events (EPEs)…"

We thank the reviewer for the suggestion. In the revised manuscript, at lines 8–9, we rephrased the sentence to read: *"The Hourly Precipitation Extremes (HPEs) are selected from this dataset…"* to clearly convey that HPEs represent the most extreme subset of the identified rainfall structures.

- Line 111: "event-based approach." As discussed earlier, this would be better described as an approach using hourly rain spatial structures. I also suggest adapting the phrase to be less generic and more explicit about how clustering helps address the limitations: "In light of these limitations, an object-based approach using a clustering technique was adopted to capture coherent hourly precipitation structures and reduce sensitivity to small-scale discrepancies between simulations and observations."

We thank the reviewer for this suggestion. In the revised manuscript, at lines 100-102, we replaced the generic term *"event-based approach"* with a more explicit formulation: *"a structure-based approach has been adopted to mitigate positioning uncertainty issues, through the use of a clustering technique"*.

- Line 146: "The event-detection … precipitation events…" The vocabulary of event throughout the text before line 164 is misleading, since line 164 specifies: "Hereafter, the term 'event' denotes the precipitation structures identified using this method." Event terminology should not appear earlier without this clarification.

We thank the reviewer for this comment. In the revised manuscript, we consistently replaced the term *"event"* with *"Hourly Precipitation Spatial Structures (HPSSs)"* throughout the text, ensuring that the terminology aligns with the definition introduced at line 179-180.

- Line 296: "is conducted on a subset of EPEs." A clearer phrasing would be: "a subset of the HOPE-X dataset." (But HOPE-X should be renamed)

We thank the reviewer for this suggestion. In the revised manuscript, the dataset has been renamed HOPSS-X (*HOurly Precipitation Spatial Structures and eXtremes*) and first introduced at line 7. Regarding the unclear phrasing, in the revised manuscript, at line 325, we now state: *"A subset of the dataset HOPSS-X is obtained"* to clearly indicate the focus on the extreme precipitation structures.

- Lines 321–322: The phrase "seasonal redistribution is likely driven by the persistence of summer-like convective activity into early autumn" is confusing, since the focus is on differences between summer and autumn. This cannot be justified by describing autumn activity as "summer-like." A phenomenon-based explanation could instead highlight that the persistence of warm sea surface conditions beneath a cooler atmosphere creates instability favorable to convection. Autumnal convection is not "summer-like". This is also reflected in Fig. 12, which shows differences in the "size" characteristics of hourly rain structures between summer and autumn.

We thank the reviewer for this insightful comment. In the revised manuscript, we replaced the confusing phrase *"summer-like"* with a phenomenon-based explanation, highlighting that the persistence of warm sea surface conditions beneath a cooler atmosphere creates instability favorable to convection, as you suggested. This clarification is provided at lines 348–350.

- Line 354: Instead of "alternating pattern," I suggest: "spatial heterogeneity in the sign of the signal" or "patterns of alternating signs."

We thank the reviewer for this suggestion. In the revised manuscript, at line 383, we replaced *"alternating pattern"* with *"spatial heterogeneity in the sign of the signal"* to improve clarity and precision.

---

## Referee Report (RR1)

*Review for revised manuscript "Hourly Precipitation Patterns and Extremization over Italy using convection-permitting reanalysis data"*

**General comments:**

The manuscript has been improved, with the authors appropriately addressing the reviewers' comments. The use of more precise terminology (e.g., "hourly precipitation patterns" instead of "events", "spatial extent" instead of "size"), a more instinctive choice of acronyms, and several redaction adjustments enhance the readability of the paper and improve the reader's understanding, particularly regarding the methodological aspects.

The contextualization of the analysis has also been strengthened. The authors now acknowledge the limitations associated with the 4-km resolution, integrate the non-extreme precipitation analysis more smoothly (through revisions to the title, abstract, and introduction), and clarify the overall purpose of the study.

Most of my previous concerns have been satisfactorily answered and incorporated. However, I reiterate that the metric N still presents a caveat. Although the authors have partially addressed this issue, it requires either further verification or more cautious phrasing in the conclusions regarding changes in extreme precipitation events (HPEs), as detailed in the specific comments.

For this reason, I recommend **minor revision** before publication.

**Specific comments:**

1. **Regarding the N metric:** My initial concern was that N could reflect both the hourly spatial density of precipitation structures and their frequency. The authors addressed this by adding, in the supplementary material (Fig. S1), an analysis of the frequency of wet hours, which reproduces the behaviour of N. These satisfactorily address the issue, but only for hourly precipitation structures, not for HPEs and not for HPE changes derived from N.
   I leave to the authors the choice to either (i) extend the wet-hour analysis to HPEs and to changes in HPEs, or (ii) explicitly state the limitations of using N to interpret HPEs behaviour (See point 2).
   In addition, in the sentence at l.204–206 *("Alternative metrics to N, such as the frequency of wet hours…")*, it should be clarified what this alternative metric is intended to test. I also do not agree that N is more "tangible" than the wet-hour metric, but this may be personal.

2. **Regarding interpretation of changes in N:** In l.384–386, the authors write: "*This suggests that changes over time are more likely associated with the frequency of HPEs rather than their intensity or spatial extent. It may also reflect the lower noise sensitivity of N compared to other indicators.*"

   This is an important statement, but in my view the methodology is not sufficient to conclude that frequency changes dominate over intensity changes in HPEs. As discussed in point 1, an increase in N could indeed reflect (i) more hours containing extremes, but it could also arise from (ii) an increase in number of structures within the 0.5 × 0.5 window during the same hour.

   The point about noise sensitivity is relevant and should be explicitly incorporated into the interpretation in that same phrase: N is less affected by noise, which makes statistical significance easier to achieve compared to peak-int and mean-int. However, both peak-int and mean-int also show increases in the selected regions (though not significant), and there are high chances that an intensification is responsible for the increase in the number of extremes.

   For these reasons, this interpretation should be rewritten with clearer limitations, as a deeper investigation would be needed to affirm that frequency is primarily affected compared to intensity in my opinion.

**Technical comments:**

1. L. 248-249 "*Before focusing on the extremes, an analysis of the overall patterns of HPSSs across the dataset is presented, providing the necessary context for the interpretation of subsequent results on extremes*". It is not straightforward that the non-extreme precipitation context is "necessary" for extreme precipitation analysis. Please clarify or rephrase.

2. L.307-308 Regarding Figure 8 "*In autumn, slightly higher intensities, ranging from 4 to 5 mm/h, cover most of the country, while lower values persist **only** in the Prealpine and Alpine regions.*" Change "only" to "mostly", as values under 4 mm/h are not exclusive to the Prealpine and Alpine regions.

3. At several points the manuscript refers to 'bias' and 'inconsistencies' in the dataset. Please clarify that these refer to findings from previous work, since the dataset is not evaluated in this study. (i) l.320-323 "*However, while MeanInt and PeakInt seasonal maps appropriately reflect higher values during the autumn and summer seasons, **they also display certain inconsistencies**. In particular, some areas exhibit an overrepresentation of convective activity during summer, which may not fully align with observed patterns. This issue will be examined in greater detail in the Discussion section 4.*" Please clarify what is meant by "inconsistencies". My

understanding is that these refer to inconsistencies in the sense of biases relative to observations. If this is the case, please specify which observational dataset they are compared to and cite the corresponding references. If it is not the case, give more precision on what you mean by inconsistencies. (ii) + l.402-403 *"Then, as described in Section 2.1, a precipitation overestimation bias is present in summer. These localized wet biases are likely due to overly active explicit convection in the model, as shown in Figure 8."* Precise what is shown in Figure 8, as it is not a bias map. Rephrase the sentence 402-403 so that it is more explicit that you are referring to bibliography here.

4. Discussion l.413-414: *"In principle, such biases could have masked decreasing trends in those areas; however, the overall spatial pattern suggests that this scenario is highly unlikely."* I do not understand how the overall spatial pattern suggests this scenario unlikely. Also be careful as you are mixing bias on annual precipitation percentages with percentages of trends in HPEs. Are they comparable? What if the bias in annual precipitation specifically comes from a misrepresentation of extremes?

---

## Author Response (AR2)

**Response to Reviewers**

**EGUSPHERE-2025-3455: "Hourly Precipitation Patterns and Extremization over Italy using Convection-Permitting Reanalysis Data"**

[answers in blue]

**Reviewer #2**: accepted as it is, but:

However, I still have some concerns regarding the Rx1hour threshold and the precipitation structures. If 11% of summer events exceed the threshold. How does this threshold compare to other more used thresholds in the literature? Eg. Rx1day, and various percentiles? The threshold of above the mean av the Rx1hour during the 37 years is intuitively very high, although justified in the reply. However, if it needs to be this high to be extreme, how does this relate to other studies mentioned in the manuscript that is also classified as extreme (for example line 94, the 90th percentile is stated as extreme). This seems inconsistent to the reader. The manuscript would be improved if the metrics were more relatable, since the structural-based approach is rather new. Some added explanations and relatability in the discussion section would improve the manuscript.

We thank Reviewer #2 for the positive assessment of the manuscript and for the constructive comments regarding the definition of extreme precipitation thresholds. These remarks are very helpful in improving the clarity and interpretability of the study.

The threshold Rx1day refers to a longer accumulation period and therefore leads to numerically larger values (in mm/day), while also capturing partially different physical phenomena. Daily maxima are not necessarily associated with short-lived convective events, which are instead the specific focus of the hourly precipitation analysis presented in this study.

Concerning percentile-based definitions of extremes, as reported in the literature review (e.g. the 90th percentile you mentioned), we acknowledge that there is no unique or universally accepted definition of "extreme" precipitation. Different definitions emphasise different aspects of the precipitation distribution and can all be legitimately referred to as extremes, despite being conceptually different and not directly comparable. Percentile-based thresholds characterise the upper tail of the full distribution. In contrast, thresholds based on annual maxima focus exclusively on the most extreme values reached over a time window (a year).

As an illustrative example, we can consider the 99th percentiles of wet-hour precipitation (hours with precipitation > 1 mm) for the four seasons (Figure below, without spatial smoothing). These statistics show that, particularly in winter, percentile-based thresholds are dominated by the large number of hours with weak precipitation, so that the few intense events contribute little to the resulting values. In contrast, a threshold derived from annual hourly maxima isolates only the most extreme value at a given location. This cuts out most of the lower values, but makes this kind of threshold conceptually different from percentile-based ones.

[Figure]

Seasonal levels associated to 99th percentile (wet hours) (1986-2022)

This does not imply that high percentiles are an incorrect way to define extremes; rather, within the context of a structure-based approach, we found a threshold based on annual maxima to be more appropriate and consistent with extreme value theory (e.g. Coles et al., 2001). Most importantly, the Rx1hour threshold is applied not at the single-grid-point level but to spatial clusters defining precipitation events. An event is classified as extreme if at least one point within the cluster (its maximum) exceeds the local threshold, effectively enlarging the set of events identified as extreme.

If the reviewer inquiry instead concerns the relationship between the mean Rx1hour threshold and percentile-based thresholds, we have prepared an additional figure (shown below) illustrating which percentile (computed among wet hours with precipitation > 1 mm) corresponds to the mean Rx1hour value (in mm/h). As noted, this percentile generally exceeds the 99th percentile and is often close to the 99.9th percentile. Nevertheless, a substantial number of events are selected because the threshold is applied to spatial clusters rather than to individual grid points.

[Figure]

Following the reviewer's suggestion, we have added a clarifying statement in the Discussion section (lines 437-439).

**Reviewer #3**: Minor Revision:

**General comments**: The manuscript has been improved, with the authors appropriately addressing the reviewers' comments. The use of more precise terminology (e.g., "hourly precipitation patterns" instead of "events", "spatial extent" instead of "size"), a more instinctive choice of acronyms, and several redaction adjustments enhance the readability of the paper and improve the reader's understanding, particularly regarding the methodological aspects. The contextualization of the analysis has also been strengthened. The authors now acknowledge the limitations associated with the 4-km resolution, integrate the nonextreme precipitation analysis more smoothly (through revisions to the title, abstract, and introduction), and clarify the overall purpose of the study. Most of my previous concerns have been satisfactorily answered and incorporated. However, I reiterate that the metric N still presents a caveat. Although the authors have partially addressed this issue, it requires either further verification or more cautious phrasing in the conclusions regarding changes in extreme precipitation events (HPEs), as detailed in the specific comments. For this reason, I recommend minor revision before publication.

We sincerely thank the Reviewer for the careful re-evaluation of the manuscript and for the positive and encouraging assessment of the revisions.

We also thank the Reviewer for reiterating the remaining concern regarding the metric N. We recognize that this aspect still warrants caution and appreciate the opportunity to further clarify and refine our interpretation. Below, we provide point-by-point responses to the specific comments, detailing the additional verifications performed and the corresponding revisions made to the manuscript.

We believe that these additional clarifications and adjustments further strengthen the robustness and transparency of the study, and we thank the Reviewer again for the constructive feedback that contributed to improving the manuscript.

**Specific comments:**
1. Regarding the N metric: My initial concern was that N could reflect both the hourly spatial density of precipitation structures and their frequency. The authors addressed this by adding, in the supplementary material (Fig. S1), an analysis of the frequency of wet hours, which reproduces the behaviour of N. These satisfactorily address the issue, but only for hourly precipitation structures, not for HPEs and not for HPE changes derived from N. I leave to the authors the choice to either (i) extend the wet-hour analysis to HPEs and to changes in HPEs, or (ii) explicitly state the limitations of using N to interpret HPEs behaviour (See point 2). In addition, in the sentence at l.204–206 ("Alternative metrics to N, such as the frequency of wet hours…"), it should be clarified what this alternative metric is intended to test. I also do not agree that N is more "tangible" than the wet-hour metric, but this may be personal.

We thank the Reviewer for the detailed comment and for the constructive suggestions regarding the interpretation of the metric *N*. We fully acknowledge the potential ambiguity of *N*, as it may reflect both the hourly spatial density of precipitation structures and their frequency.

Following the Reviewer's recommendation, we extended the frequency analysis also to HPEs:

[Figure]

This additional analysis shows that, for extreme precipitation, the spatial and seasonal patterns of the frequency of occurrence are highly consistent with those obtained using *N* *(see Figure 10 of the manuscript),* with no substantial differences that would suggest a dominant role of frequency changes over spatial density. The same conclusion applies to the temporal changes, as trends in HPE wet-hour frequency closely mirror those derived from *N*:

[Figure]

As these additional figures do not provide further independent insight beyond what is already conveyed by *N*, we opted not to include them in the Supplementary Material. However, we now explicitly state in the Method section 2.5 (lines 239-240) that the behaviour illustrated in Fig. S1 for hourly precipitation structures also applies to HPEs and to their temporal trends.

We have also revised the sentence at lines 204-207, specifying that the purpose of the alternative metric based on wet-hour frequency is precisely to disentangle the respective contributions of the hourly spatial density of precipitation structures and their frequency of occurrence.

2.  Regarding interpretation of changes in N: In l.384–386, the authors write: "This suggests that changes over time are more likely associated with the frequency of HPEs rather than their intensity or spatial extent. It may also reflect the lower noise sensitivity of N compared to other indicators." This is an important statement, but in my view the methodology is not sufficient to conclude that frequency changes dominate over intensity changes in HPEs. As discussed in point 1, an increase in N could indeed reflect (i) more hours containing extremes, but it could also arise from (ii) an increase in number of structures within the 0.5 × 0.5 window during the same hour.

We thank the Reviewer for this important observation and for highlighting the need for caution in interpreting changes in N. We agree that, in principle, an increase in N may arise from (i) an increase in the number of hours containing HPEs and (ii) an increase in the number of HPEs occurring within the same spatial window during a single hour.

However, case (i) result the dominant mechanism, since the analysis of wet-hour frequency for HPEs shows spatial, seasonal, and temporal patterns that closely mirror those of N, including comparable trends, indicating that changes in N primarily reflect an increasing number of hours containing extreme precipitation. Moreover, we believe that an increase in the number of HPEs exceeding the threshold within the same spatial window and at the same hourly time step (i.e., simultaneous events) does not represent a methodological issue in itself. When more than one HPE is counted within a window at a given hour, these correspond to spatially disjoint structures and therefore to distinct precipitation events, rather than to multiple counts of the same event. As such, their contribution to N reflects a genuine increase in the number of extreme precipitation structures, not an artificial inflation of the metric.

The point about noise sensitivity is relevant and should be explicitly incorporated into the interpretation in that same phrase: N is less affected by noise, which makes statistical significance easier to achieve compared to peak-int and mean-int. However, both peak-int and mean-int also show increases in the selected regions (though not significant), and there are high chances that an intensification is responsible for the increase in the number of extremes. For these reasons, this interpretation should be rewritten with clearer limitations, as a deeper investigation would be needed to affirm that frequency is primarily affected compared to intensity in my opinion.

We have revised the sentence at lines 388-391 to explicitly incorporate this interpretation, clarifying that an intensification of extremes—while not statistically significant at the hourly scale—can lead to an increased number of HPEs ( more HPSSs exceeding the threshold). This results in a clearer and more robust signal in N, allowing statistically significant trends in HPE occurrence to emerge.

**Technical comments:**

1. L. 248-249 "Before focusing on the extremes, an analysis of the overall patterns of HPSSs across the dataset is presented, providing the necessary context for the interpretation of subsequent results on extremes". It is not straightforward that the non-extreme precipitation context is "necessary" for extreme precipitation analysis. Please clarify or rephrase.

We thank the Reviewer for this comment and agree that the term "necessary" may be too strong. The analysis of non-extreme precipitation patterns is not strictly required for the study of extremes; rather, it is intended to be supportive and informative, helping to better contextualize and interpret the results on extreme precipitation. We have revised the sentence at lines 250-251 accordingly to reflect this distinction more clearly.

2. L.307-308 Regarding Figure 8 "In autumn, slightly higher intensities, ranging from 4 to 5 mm/h, cover most of the country, while lower values persist only in the Prealpine and Alpine regions." Change "only" to "mostly", as values under 4 mm/h are not exclusive to the Prealpine and Alpine regions.

We thank the Reviewer for this suggestion. The text at line 311has been revised accordingly, replacing "only" with "mostly" to accurately reflect that values below 4 mm/h are not exclusive to the Prealpine and Alpine regions.

3. At several points the manuscript refers to 'bias' and 'inconsistencies' in the dataset. Please clarify that these refer to findings from previous work, since the dataset is not evaluated in this study:

   (i) l.320-323 "However, while MeanInt and PeakInt seasonal maps appropriately reflect higher values during the autumn and summer seasons, they also display certain inconsistencies. In particular, some areas exhibit an overrepresentation of convective activity during summer, which may not fully align with observed patterns. This issue will be examined in greater detail in the Discussion section 4." Please clarify what is meant by "inconsistencies". My understanding is that these refer to inconsistencies in the sense of biases relative to observations. If this is the case, please specify which observational dataset they are compared to and cite the corresponding references. If it is not the case, give more precision on what you mean by inconsistencies.

We thank the Reviewer for this observation. We agree that "inconsistencies" was not precise, and we have revised the text at lines 324-326 to explicitly refer to biases relative to observations, citing previous work where the dataset was compared with observational data.

   (ii) l.402-403 "Then, as described in Section 2.1, a precipitation overestimation bias is present in summer. These localized wet biases are likely due to overly active explicit convection in the model, as shown in Figure 8." Precise what is shown in Figure 8, as it is not a bias map. Rephrase the sentence 402-403 so that it is more explicit that you are referring to bibliography here.

We thank the Reviewer for this comment and agree that Figure 8 does not represent a bias map, but rather shows the spatial patterns of HPSSs seasonal mean intensities. We have revised the sentence at lines 407-409 to clarify that the overestimation in summer precipitation refers to findings reported in previous studies, rather than to Figure 8 directly.

4. Discussion l.413-414: "In principle, such biases could have masked decreasing trends in those areas; however, the overall spatial pattern suggests that this scenario is highly unlikely." I do not understand how the overall spatial pattern suggests this scenario unlikely. Also be careful as you are mixing bias on annual precipitation percentages with percentages of trends in HPEs. Are they comparable? What if the bias in annual precipitation specifically comes from a misrepresentation of extremes?

We thank the Reviewer for this careful observation. The reason for that statement is that Figure 12 shows that the majority of significant and non-significant points display an increasing tendency of HPEs occurrences. Based on this overall spatial pattern, it appears unlikely that extensive areas could instead exhibit an opposite signal that is subsequently masked by model biases. Also, we agree that the comparison between biases in annual precipitation percentages and percentages of trends in HPEs involves distinct quantities and is therefore not strictly comparable numerically. Here, the intention was qualitative, to provide an idea of the magnitude of potential impacts and to illustrate that, even if some bias could influence trends, this effect is likely to be small. We have revised the text at lines 418-420 to clarify this interpretation and to emphasize the qualitative nature of the comparison, acknowledging that the two quantities refer to different temporal intervals and precipitation characteristics.